# Surface Topography of PVD Hard Coatings

Peter Panjan [1,*], Aljaž Drnovšek [1], Nastja Mahne [1,2], Miha Čekada [1,2] and Matjaž Panjan [1]

1   Jožef Stefan Institute, Jamova 39, 1000 Ljubljana, Slovenia; aljaz.drnovsek@ijs.si (A.D.);
    nastja.mahne@ijs.si (N.M.); miha.cekada@ijs.si (M.Č.); matjaz.panjan@ijs.si (M.P.)
2   Jožef Stefan International Postgraduate School, Jamova 39, 1000 Ljubljana, Slovenia
*   Correspondence: peter.panjan@ijs.si

**Abstract:** The primary objective of this study was to investigate and compare the surface topography of hard coatings deposited by three different physical vapor deposition methods (PVD): low-voltage electron beam evaporation, unbalanced magnetron sputtering and cathodic arc evaporation. In these deposition systems, various ion etching techniques were applied for substrate cleaning. The paper summarizes our experience and the expertise gained during many years of development of PVD hard coatings for the protection of tools and machine components. Surface topography was investigated using scanning electron microscopy (SEM), atomic force microscopy (AFM), scanning transmission electron microscopy (STEM) and 3D stylus profilometry. Observed similarities and differences among samples deposited by various deposition methods are discussed and correlated with substrate material selection, substrate pretreatment and deposition conditions. Large variations in the surface topography were observed between selected deposition techniques, both after ion etching and deposition processes. The main features and implications of surface cleaning by ion etching are discussed and the physical phenomena involved in this process are reviewed. During a given deposition run as well as from one run to another, a large spatial variation of etching rates was observed due to the difference in substrate geometry and batching configurations. Variations related to the specific substrate rotation (i.e., temporal variations in the etching and deposition) were also observed. The etching efficiency can be explained by the influence of different process parameters, such as substrate-to-source orientation and distance, shadowing and electric field effects. The surface roughness of PVD coatings mainly originates from growth defects (droplets, nodular defects, pinholes, craters, etc.). We briefly describe the causes of their formation.

**Keywords:** topography; PVD coating; ion etching; sputtering; 3D stylus profilometry; atomic force microscopy; scanning electron microscopy; focused ion beam

## 1. Introduction

Interactions between solid surfaces depend not only on the properties of the materials in contact but also on the topography of the surfaces [1]. Therefore, the surface topography can have a considerable impact on many material functional properties (e.g., the ability to adhere to another material, optical properties, friction, wear). This applies not only to bulk material but also to all kinds of coatings, including those prepared by physical vapor deposition methods. Their topography is strongly dependent both on the substrate topography and on the topography induced by the coating deposition process. While the former depends on the substrate preparation steps before the coating process, e.g., grinding or polishing, the latter mainly depends on the ion etching step before deposition as well as on the formation of growth defects during the deposition process.

The surface topography of PVD hard coatings is an important factor influencing its tribological performance under sliding contact conditions [2,3]. Namely, the real contact area strongly depends on the roughness of the interacting surfaces. The reduction in contact area in the case of a rough surface affects friction, adhesion, wear and other tribological phenomena. However, the real area of contact is not only a function of surface

roughness, but also a function of the applied normal load. Thus, in the case of an elastic contact, it increases with normal load. A sliding of two surfaces does not always generate constant friction force, but the motion changes periodically between adhesion and sliding (i.e., the stick-slip phenomenon). Rougher coating surfaces cause higher friction and wear because of abrasive and ploughing effects. Because of a smaller real contact area in the case of a rough surface, a high surface pressure at contact spots also increases the tendency for crack initiation and the risk of fatigue-related damage. On the other hand, a very smooth coating surface contact may increase the adhesion between the surfaces, and therefore, the material transfer in the contact will be more pronounced [4]. However, a certain degree of roughness can also be useful; for example, when oil is used, it allows oil retention in the sliding contact area [5,6]. Particularly, surface irregularities (e.g., pores, crevices) on rough sliding surfaces can store lubricant and supply it to the interface. At these sites, the wear particles can also be trapped, which can significantly reduce abrasive wear. Thus, the challenge is to find the optimal surface roughness for the contacting surfaces to achieve optimal tribological performance of the coating.

Wettability is the next example of the coating surface property, which highly depends not only on its chemical composition but also on its topography. Applications where solid surface wettability plays a crucial role include contact lenses, super-hydrophilic surfaces, self-cleaning (lotus effect), nonstick surfaces, biofilm growth and body implants. Generally, the wettability of solid surfaces is evaluated by contact angles at the liquid droplet/solid surface/air interfaces. The balance at the three-phase contact is given by the Young equation, where it is assumed that the surface is chemically homogeneous and topographically ideally smooth. However, this is not true in the case of a real surface. Instead of one equilibrium contact angle, a range of contact angles exist, because the actual contact angle is the angle between the tangent to the liquid–vapor interface and the actual local solid surface. When using water as a liquid phase, a surface is defined as hydrophilic for a contact angle less than 90° (the liquid will subsequently spread over the surface) and hydrophobic for a contact angle greater than 90°.

The surface topography and surface roughness of the orthopedic and dental implants (in addition to the chemical composition) have a decisive influence on their integration and biological response in soft and hard tissues [7–10]. Several biocompatible wear-resistant and antimicrobial PVD coatings have been developed, which can improve selected surface properties (e.g., hardness, wettability, elastic strain, friction coefficient, wear) of implants (e.g., hip and knee prostheses) [11]. It was established that surfaces containing complex, rough, textured, and porous topographical features stimulate the protein adsorption and subsequent cell adhesion on implant materials and the formation of the extracellular matrix [10]. This is mainly because complex topographical features on the surfaces provide, in comparison with smooth ones, more surface area for interaction with the proteins and surrounding physiological environment. In recent years, surface topography has also been found to substantially influence the interaction between bacteria and surfaces [9]. In general, a large surface area with a rough surface promotes bacterial adhesion, while surfaces with the specific micro- or nanoscale surface features present enhanced antibacterial properties, either by preventing bacterial adhesion or by inactivating (killing) the adherent bacteria [9]. However, smooth surfaces are desired for medical devices that are directly exposed to blood, such as cardiovascular stents, to minimize clotting and restenosis. Recently, a lot of attention has been paid to surface topography modifications of the implant surface, particularly in the nanoscale regime, and the use of biocompatible coatings, in order to mimic the surrounding biological environment as well as to prevent the infection of tissue [10].

Optical coatings are yet another example of PVD coatings, where the topography of the coating surface has a major effect on their functionality [12]. The light reflected from an imperfect optical surface consists of a specular reflected component and a diffuse reflected component. The most significant sources of light scattering are the surface roughness and growth defects (e.g., nodular defects, pinholes). In general, the scattered light is affected

only by those features that are of the same size or larger than the wavelength of light. Diffuse scattered light degrades the performance of high-precision optical systems for several reasons: (i) it reduces optical throughput (since some of the scattered light will not even reach the focal plane), (ii) it causes a reduction in contrast (signal-to-noise ratio) and (iii) the small-angle scatter causes a decrease in resolution (image blur) [13].

Although the topography of coatings strongly affects the performance of PVD coatings, literature systematically addressing this subject is scarce. Authors most often cite only some general data, such as surface roughness, while a deeper view of these issues is missing. Only a few studies are devoted to the influence of mechanical pretreatment and ion etching on the topography of PVD coatings [14,15]. In this paper, we limit ourselves to the topography of PVD hard coatings deposited on different substrates by various PVD methods used in industrial production. There are several significant differences between these methods both in the ion etching and the deposition steps. Therefore, we describe and analyze all coating preparation steps that can affect the topography of the coating. This includes selection of substrate and coating material, mechanical and chemical pretreatment, ion etching, deposition methods and deposition parameters. Such studies are particularly important for the reduction in the surface density of artefacts such as growth defects.

## 2. Materials and Methods

### 2.1. PVD Processes

All coatings were prepared in industrial batch-type deposition systems. A more extensive description of the four deposition processes is given below, while the essential process parameters and schematic drawings for each deposition method are given in Table 1 and Figure 1.

*Low-voltage electron beam evaporation (or thermionic arc evaporation) system BAI 730* (BAI, Balzers, Vaduz, Liechtenstein) was used for deposition of TiN single layer coatings (Figure 1a). This system consists of a thermo-emissive cathode (filament), a crucible (evaporation source) connected to a low-voltage supply, and an auxiliary anode (around the target) [16]. The plasma (thermionic arc discharge), created between the ionization chamber and auxiliary anode, is used for the heating, etching and evaporation steps. A great advantage of this method is that the substrates can be immersed in a very dense plasma. Prior to deposition, the substrates are heated by applying a positive voltage to the substrate table, attracting electrons from the plasma and heating the substrates up to 450 °C. Then, the surface of substrates is cleaned by ion etching for 15 min. Argon ions are drawn out from the arc discharge and accelerated towards the substrates using a voltage of −200 V. To regulate the etching intensity, the substrate voltage can be adjusted independently. The low inert gas pressure (e.g., 0.1 Pa) ensures that the mean-free-path length is much longer (e.g., 5 cm) than the structural details of the substrates, thus providing excellent penetration. The plasma boundaries perfectly follow the contours of complex-shaped substrates because, according to the Child–Langmuir law, the width of the dark space is less than 1 mm in a high-density plasma (e.g., >1 mA/cm$^2$). Control of the etching process is easy and there are fewer problems with mixed loads. After the etching step, the crucible is made the anode of the arc discharge. The evaporation material held in the water-cooled copper anodic crucible is heated by electrons impinging on the crucible from the thermionic arc. During deposition, the bias voltage on the substrates is −125 V. The ion flux to the substrates is approximately proportional to the film deposition rate because of the high amount of coating material vapor ionizes in the plasma. In order to improve the coating uniformity, the rotating cylindrical substrate holders are placed concentrically around the crucible. Additionally, the crucible moves vertically during deposition to further improve the coating uniformity. The deposition rate of TiN hard coating is about 50 nm/min (for 2-fold rotation of the substrate).

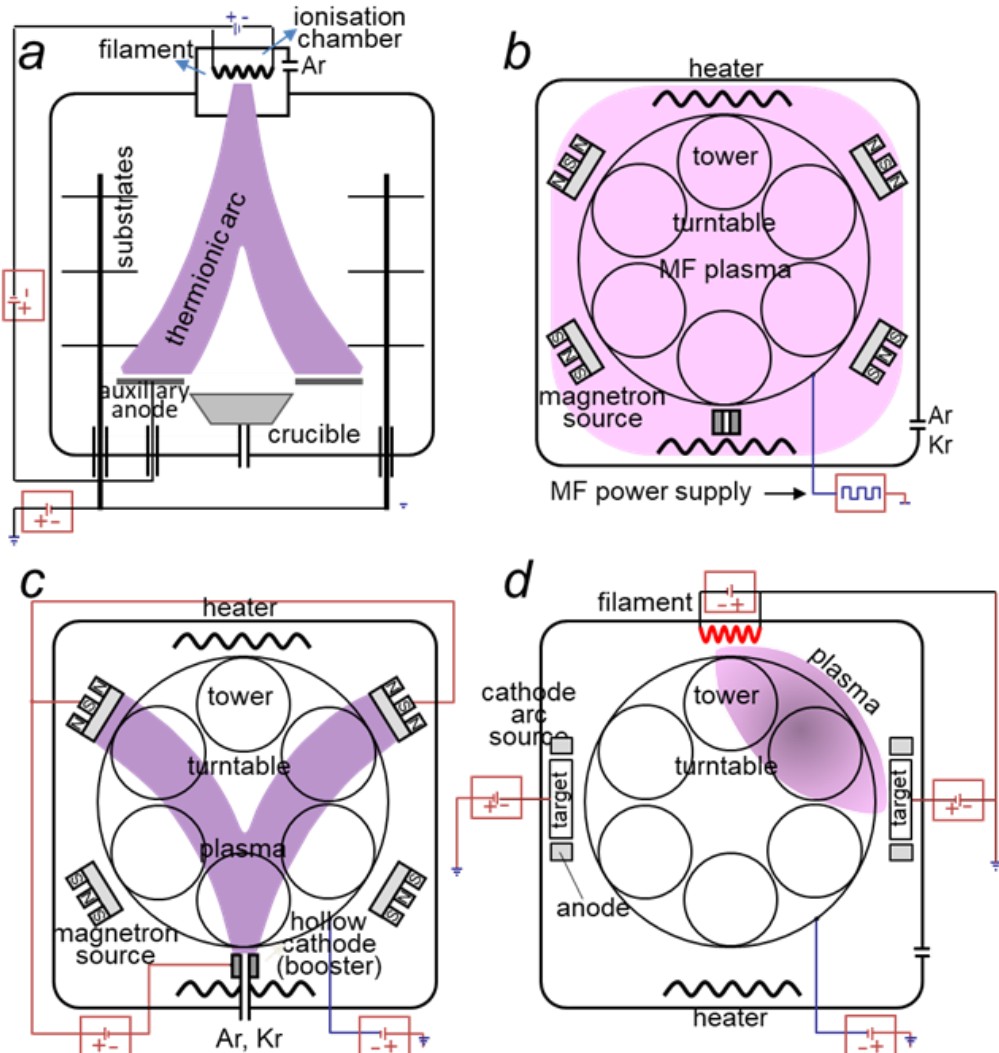

**Figure 1.** Schematic drawings of different deposition methods and ion etching modes used for the preparation of PVD hard coatings (the approximate area of plasma during the ion etching is marked by purple color) [17]: (**a**) side view of ion etching (DC bias) in low-voltage electron beam evaporation system BAI 730 (BAI); (**b**) top view of RF etching in magnetron sputtering system CC800/7 (CC7) and MF etching in CC800/9 sinOx ML (CC9); (**c**) top view of etching with hollow cathode plasma source (DC bias) in magnetron sputtering system CC9; (**d**) top view of ion etching (DC bias) cathodic arc deposition AlPocket (AIP).

A modified *magnetron sputter deposition system CC800/7* (CC7, CemeCon, Würselen, Germany) was used for depositions of TiAlN and TiN hard coatings (Figure 1b). This system is equipped with four 8 kW unbalanced magnetron sources, arranged in the corners of a chamber and operating in DC mode. The dimensions of an individual source are 200 mm × 88 mm. They are not positioned at the same height but at different vertical positions, which improves coating uniformity. This system utilizes infrared heaters for heating the substrates prior to the ion etching. During substrate cleaning, an RF bias is applied (maximum RF power 2 kW), while the etching time is 85 min. During deposition, the total operating pressure is maintained at 0.75 Pa, with the flow rates of nitrogen, argon and krypton being 100, 160 and 110 sccm, respectively. High-purity gases (99.998%) are used. Additional plasma is generated between the magnetron and the auxiliary anode–hollow cathode (such "booster" module is used during the deposition to enhance the plasma density). A DC bias of −95 V is applied to the substrates. The deposition time is 135 min. After this time, the deposition process is interrupted for another intermediate

ion etching (for 55 min at the same conditions as in the initial substrate etching step). This is followed by an additional deposition of a coating (deposition time is 30 min). The intermediate etching creates new nucleation sites for the subsequently deposited nitride coating, resulting in a fine-grained and less porous microstructure of the top layer.

**Table 1.** Deposition methods and process parameters used for the preparation of PVD hard coatings. The deposition rates and ion current density are averages since all deposition techniques exhibited both spatial and temporal variations during a given deposition run as well as from one run to another run (depending on batching material).

| | **Deposition System** | **BAI 730** | **CC800/7** | **CC800/9 sinOx ML** | **AIPocket** |
|---|---|---|---|---|---|
| **preheating** | heating method | Electron bombardment | Infrared heating | infrared heating | infrared heating |
| | preheating temperature (°C) | 450 | 450 | 450 | 450 |
| **etching** | etching mode | DC | RF | MF/booster | pulsed DC |
| | type of ions | Ar | Ar + Kr | Ar + Kr | Ar |
| | negative substrate etching voltage (V) | 200 | 200 | 650/200 | 300/400 |
| | etching time (min) | 15 | 85 + 55 ** | 15/60 | 45 |
| **deposition** | deposition method | low-voltage electron beam evaporation | Magnetron sputtering | magnetron sputtering | cathodic arc evaporation |
| | temperature (°C) | 450 | 450 | 450 | 450 |
| | working gas | Ar + $N_2$ | Ar + Kr + $N_2$ | Ar + Kr + $N_2$ | $N_2$ |
| | pressure of working gas (Pa) | 0.2 | 0.75 | 0.66 | 4 |
| | deposition time (min) | 80 | 165 | 200 | 45 |
| | negative substrate bias voltage (V) | 125 | 95 | 90 | 70 |
| | average deposition rate * (nm/min) | 50 | 20 | 20 | 85 |
| | average substrate current density (mA/cm$^2$) | 3–5 | ~2 | ~2.5 | - |

* For twofold rotation of substrates, ** intermediate etching.

The magnetron sputter deposition system CC800/9 sinOx ML (CC9, Cemecon, Würselen, Germany) was used for the deposition of different nanostructure (nl-nanolayer, nc-nanocomposite) hard coatings (nl-TiAlN/CrN, nl-TiAlN/TiN, nl-AlTiN/TiN, nc-TiAlN/TiSiN/TiAlSiN, nc-TiAlCrSiN) as well as for the deposition of TiAlN/a-CN and TiAlN/Al$_2$O$_3$ double layer hard coatings (Figure 1c). In this system, the chamber is equipped with four rectangular magnetron sputtering cathodes (500 mm × 88 mm) which can operate in DC (TiAlN, TiN, CrN, AlTiN, nc-TiSiN, nl-TiAlSiN, nc-TiAlCrSiN, a-CN) or pulsed DC modes (Al$_2$O$_3$). The turntable can provide up to threefold rotation of the substrates. Prior to the depositions, the chamber is evacuated to a base pressure of 3 mPa, and heated to around 450 °C. In the next step, the substrates are ion-etched for 55 min in a mid-frequency plasma (Ar and Kr gas mixture, 240 kHz, duty cycle 1600 ns), with a bias voltage of 650 V, applied to the substrate table. An additional etching option, primarily used for round cutting tools, is so-called "booster" etching, where the working gas is injected through upper and lower "booster" etch nozzles (i.e., the hollow cathode), where intensive ionization of the working gas (Ar, Kr) occurs. The plasma is created between the magnetron cathodes that are facing opposite to the "booster" (see Figure 1c). Such additional discharge enhances the plasma density and thus the intensity of the

etching process. Nitride-based coatings are deposited in a mixture of argon (160 sccm), krypton (110 sccm) and nitrogen (80 sccm), at a total pressure of 0.66 Pa. A DC bias of $-90$ V is applied to the substrates. The "booster", which is active also during the deposition process, enhances the potential gradient in the vacuum chamber so that the ionization degree of the sputtered atoms is increased to more than 50% [18]. The drawback of the "booster" etching is associated with shadowing, where some parts of the tool with complex geometries can be less exposed to ion etching.

Another series of the coatings was deposited by *cathodic arc evaporation deposition system AIPocket* (AIP, KCS Europe GmbH, Monschau, Germany). This system uses a technology called "super fine cathode" (SFC) [19]. SFC is a magnetically controlled cathodic arc source, which enables the deposition of low-stress, relatively smooth and thick coatings. In this system, prior to the deposition, the substrates are heated up to 450 °C and then they are cleaned by argon ion etching process. The ion etching in the AIP deposition system is also based on the auxiliary plasma (Filament-Assisted Pulse Etching or FAPE™ technology). In this system, the plasma is generated between a resistance-heated tungsten filament and two nearby cathodes (Figure 1d). Argon ions are drawn out from the arc discharge and accelerated to the substrate using a (pulsed) voltage of $-400$ V. The voltage and the current between the filament and the cathodes are 65 V and 6 A, respectively. For the coating deposition process, four targets are employed, and a power of 3 kW (20 V/150 A) is applied to each of the four targets. The deposition pressure is 4 Pa, while a bias voltage of 70 V is applied to the substrate table. The deposition time is 45 min, which results in 4 μm thick coatings on substrates mounted on a turntable that has the possibility of threefold rotation.

### 2.2. Substrate Materials

Six different substrate materials were used in this work: conventional tool steels (M2, D2, H11), powder metallurgical (PM) steel (ASP30), stainless steel (SS 316L) and cemented carbide (HM). The round substrates (3 mm in thickness, 18 mm in diameter) were ground and polished using 1 μm diamond paste for the final polish. Before deposition, the samples were degreased and cleaned in an industrial sized automated ultrasonic cleaning line. The first step was ultrasonic degreasing in de-ionized water with a degreaser (pH ~11) to remove surface impurities (cleaning time 15 min), followed by ultrasonic rinsing in deionized water and then drying in pure hot air. The coatings were deposited on all six different substrate materials with 1-fold, 2-fold and 3-fold rotations. Most substrates were mounted in the middle vertical position, while some of them were positioned at various heights in the vacuum chamber. Most substrates were mounted in the middle of the vertical position, while some of them were positioned at the bottom and top positions of the substrate holder.

### 2.3. Analytical Methods

Different analytical techniques were used to characterize the surface morphology and surface roughness as well as to look at individual features on the surface. Surface topography was observed with an optical microscope (OM) Axio CSM 700 (Carl Zeiss Microscopy, Oberkochen, Germany). However, the lateral and depth resolution of the optical microscope is not good enough for detailed analysis of micrometer-sized defects. Additionally, in the optical image, all types of growth defects are seen as black dots and therefore, it is impossible to distinguish between protrusions and craters. The most common technique for the study of substrate and coating morphology is scanning electron microscopy (SEM). In our study, a JEOL JSM-7600F SEM microscope (JEOL, Tokyo, Japan) was used. The applicability of SEM microscopy is limited due to relatively low-depth resolution. Therefore, atomic force microscope Solver PRO (AFM,) (NT-MDT Spectrum Instruments, Moscow, Russia) was employed to study the surface topography on a nano level. The use of an AFM microscope is limited due to the small scan area (of about 50 μm × 50 μm) and is therefore not appropriate for the study of growth defects, because its density is rather low (a few hundred defects/mm$^2$). The most suitable method for this

purpose is a 3D stylus profilometry [20]. In the scanning mode, the stylus profilometer can give a 3D image of the surface on a large scan area (from several hundreds of micrometers square to several millimeters square) with all the micrometer-sized details. The Bruker Dektak XT stylus profilometer was used in this study.

The microstructure and the coating morphology were studied using fracture cross-sections examined in the SEM microscope. Cross-sections for SEM investigations were also prepared by focused ion beam techniques (FIB) using a FIB source integrated into the Helios Nanolab 650i field emission scanning electron microscope (FEI, Amsterdam, The Netherlands). Detailed microstructural analysis was performed by JEOL ARM 200 CF transmission electron microscope (TEM). The specimen for TEM characterization was prepared with a Helios NanoLab 600i focused ion beam system using the standard lift-out technique. SEM and TEM images were recorded using the ion beam and the electron beam.

## 3. Results and Discussion

In general, the topography of a PVD coating may originate from three different contributions [21]: (a) topography of the substrate surface, (b) intrinsic coating micro-topography or (c) growth defects forming during the deposition process.

### 3.1. Topography of the Substrate Surface

Substrate surface preparation is an integral part of any PVD film deposition process. The topography of the surface of an engineering material is determined by the method of its processing and the nature of the material itself. In general, the substrate surface topography may originate from mechanical pretreatment (grinding, blasting, polishing) and ion etching [3,14].

### 3.1.1. Substrate Irregularities Induced by Mechanical Pretreatment

Mechanical pretreatment of tool materials (e.g., tool steels, cemented carbides) usually includes grinding, blasting and polishing. Such pre-treatment significantly smooths the substrate surface, but it also creates different micrometer-size irregularities [22]. Thus, grinding, even if performed carefully, creates various irregularities of different shapes (such as scratches, grooves, ridges, pits) that are produced by the abrasive SiC particles in the grinding paper. The influence of mechanical preparation on the topography of the substrate surface is described in more detail in our recent review paper [17].

During polishing, which is the next step in the mechanical treatment of the substrates, protrusions are generated. These are formed due to hardness difference between the different phases in the substrate material. Namely, the removal rate depends on the hardness of the individual phase. Therefore, the harder phases that are more resistant against polishing will protrude from the surface (Figure 2) [14]. In such cases, the surface topography reflects the microstructure of the substrate material. As it can be seen in Figure 2, both the ASP30 and D2 tool steel substrates have higher roughness in comparison to H11 steel. After polishing the D2 and ASP30 tool steel substrates, shallow protrusions (about 8 nm in height) appeared at the sites of carbide inclusions, while no such protrusions are observed in the case of H11 tool steel. Namely, the amount and type of carbides depend on the carbon content and the quantity of carbide-forming elements (chromium, molybdenum, vanadium and tungsten, for example). The carbon content in H11 steel is much lower than in ASP30 and D2 steel. Therefore, the carbides in H11 steel are also much smaller and distributed uniformly in the steel matrix. The homogeneity of the microstructure, the level of purity, the size and distribution of carbides and other hard constituents in the steel matrix are the parameters that influence the polishability of tool steel substrates [22].

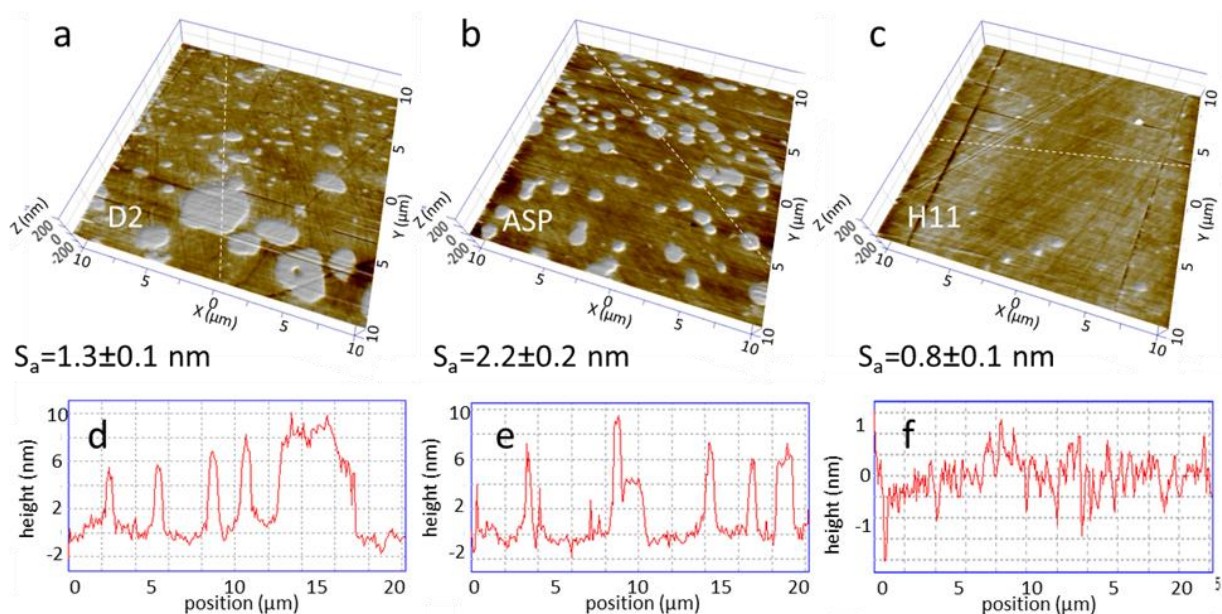

**Figure 2.** AFM images and surface roughness ($S_a$) of bare D2 (**a**), ASP30 (**b**) and H11 (**c**) tool steel substrates after polishing. The scan area was 20 μm × 20 μm, while the z-scale is 200 nm. Figures (**d**–**f**) show typical roughness profiles along the dashed lines in figures (**a**–**c**). Pay attention to the different z-scales.

### 3.1.2. Substrate Irregularities Induced by Ion Etching

In this paper, we examine the substrate irregularities induced by ion etching in typical industrial PVD deposition systems. The ion etching procedures are in part described in the experimental section (see schemes in Figure 1). Here, we provide additional details for the four deposition systems used in this work: magnetron sputtering systems (CC7, CC9), cathodic arc evaporation system (AIP) and the low-voltage electron beam evaporation system (BAI). In these four deposition systems, two different concepts of inert gas (Ar or Ar + Kr) ion etching are used. The first concept of ion substrate etching is based on the plasma generated by applying oscillating voltage between the substrate and the electrically grounded vacuum chamber. Such ion etching can only operate if high-frequency oscillatory voltage is applied. A radio frequency (RF) with 13.56 MHz and middle frequency (MF) with 240 kHz are used in CC7 and CC9 deposition systems, respectively. An advantage of this type of ion etching is that it results in a more-or-less uniform etching of all substrates in the deposition system. However, the disadvantage is that it does not allow independent control of ion densities and energies. The etching efficiency depends on the voltage frequency and amplitude.

In the second concept of substrate ion etching, an auxiliary plasma source is used (BAI, "booster" etching in CC9, AIP). The auxiliary plasma is normally spatially confined and therefore, not all substrates are immersed in the plasma at the same time. In order to achieve a uniform etching, the substrates need to rotate past the dense plasma. However, the main advantage of auxiliary plasma is that the ion density can be controlled by the plasma source, while the ion energy is controlled by the bias potential on the substrates. Furthermore, substrates can be biased either by continuous, pulsed, or oscillatory potential, which offers even greater control over the etching parameters. Although continuous substrate bias provides the most intense substrate etching, a pulsed or oscillatory substrate potential is still used in many cases because it enables the removal of native oxide and other non-conductive contaminates.

The above-mentioned concepts of ion etching are based on the use of inert gas ions. In this case, implanted ions can cause tensile stresses or even the formation of an amorphous zone [23]. This is not the case if metal ions are used. Metal ions can be generated in the deposition system with cathodic arc or high-power impulse magnetron sputter (HIPIMS) sources. However, a disadvantage in the case of arc discharge is the contamination of the

substrate surface with the droplets of pure metal emitted by the cathode spots. This problem can be completely eliminated if HIPIMS discharge is used to generate the metal ions.

Ion bombardment changes the microchemistry of the substrate, its surface topography and microstructure of the near-surface layer. The ion etching results in the removal of native oxides and other contaminants, increased density of nucleation sites and chemical activation of the surface. All these changes affect not only the adhesion of the coating but also its growth. The etching efficiency depends on the etching method (Figure 3), etching time, the geometry of the substrate, substrate material (Figures 4–6), current density, ion energy, rotation mode of the substrate (Figure 7), plasma homogeneity and other parameters. The efficiency of ion etching can be determined by observing the selected area on the substrate surface under optical, SEM and AFM microscopes or by making a 3D profile image (Figure 8).

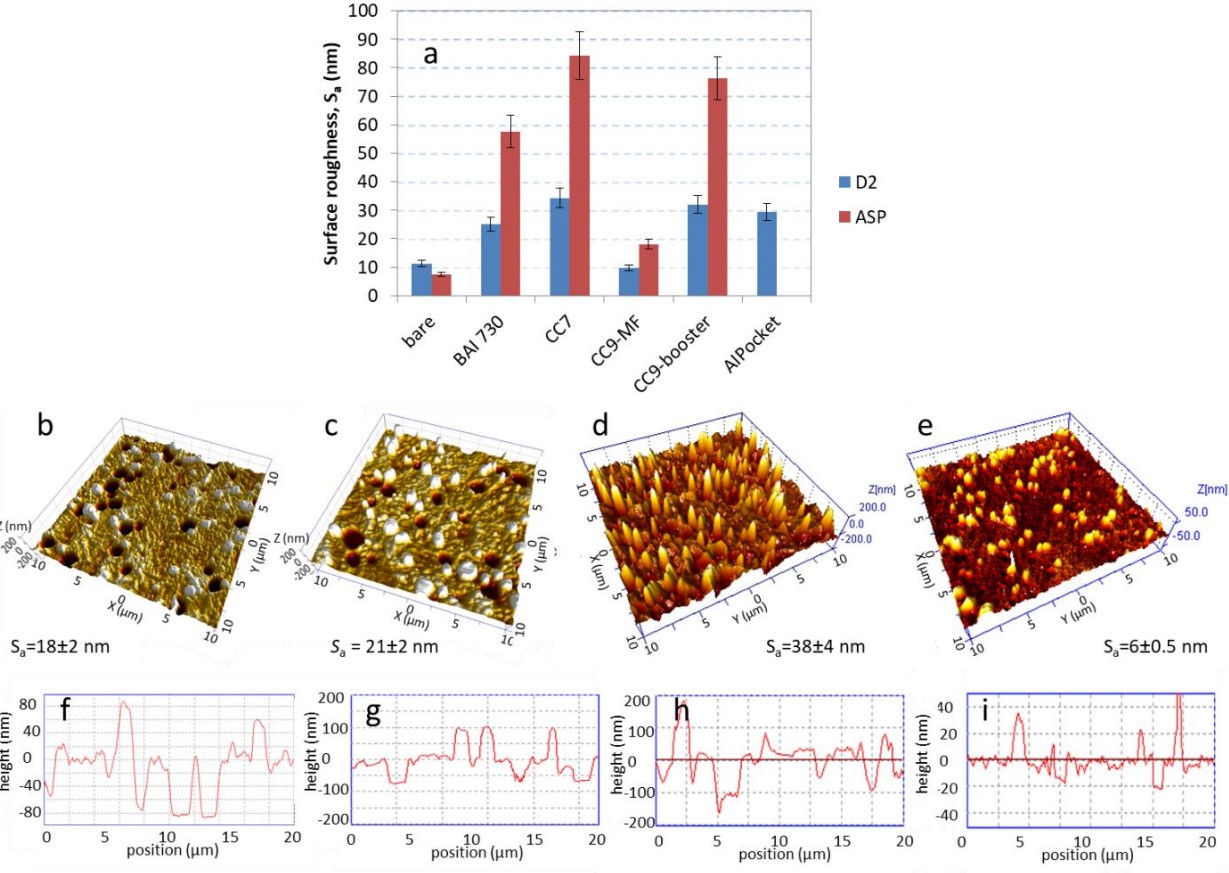

**Figure 3.** (**a**) 3D surface roughness ($S_a$) (scan area was 300 μm × 300 μm) of D2 and ASP tool steel substrates after ion etching in different industrial deposition systems. AFM images (**b**–**e**), surface roughness ($S_a$) and line profiles (**f**–**i**) of the ASP substrates (the scan area was 20 μm × 20 μm) after ion etching in: BAI 730 (**b**,**f**), CC7 (**c**,**g**), MF + booster etching in CC9 (**d**,**h**) and MF etching in CC9 (**e**,**i**). Pay attention to the different z-scales on AFM images and height scale on line profiles.

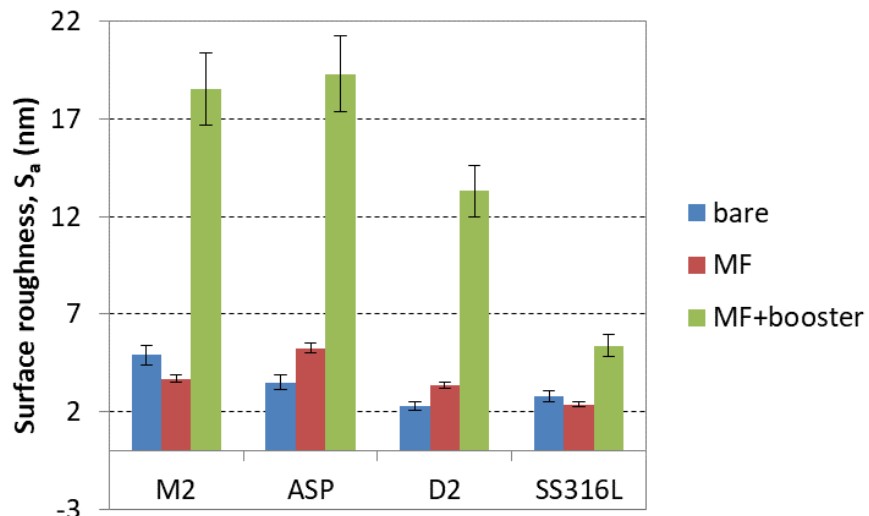

**Figure 4.** AFM surface roughness of different substrates (M2, ASP, D2, SS 316L) before and after MF and MF + booster ion etching. The scan area was 20 μm × 20 μm.

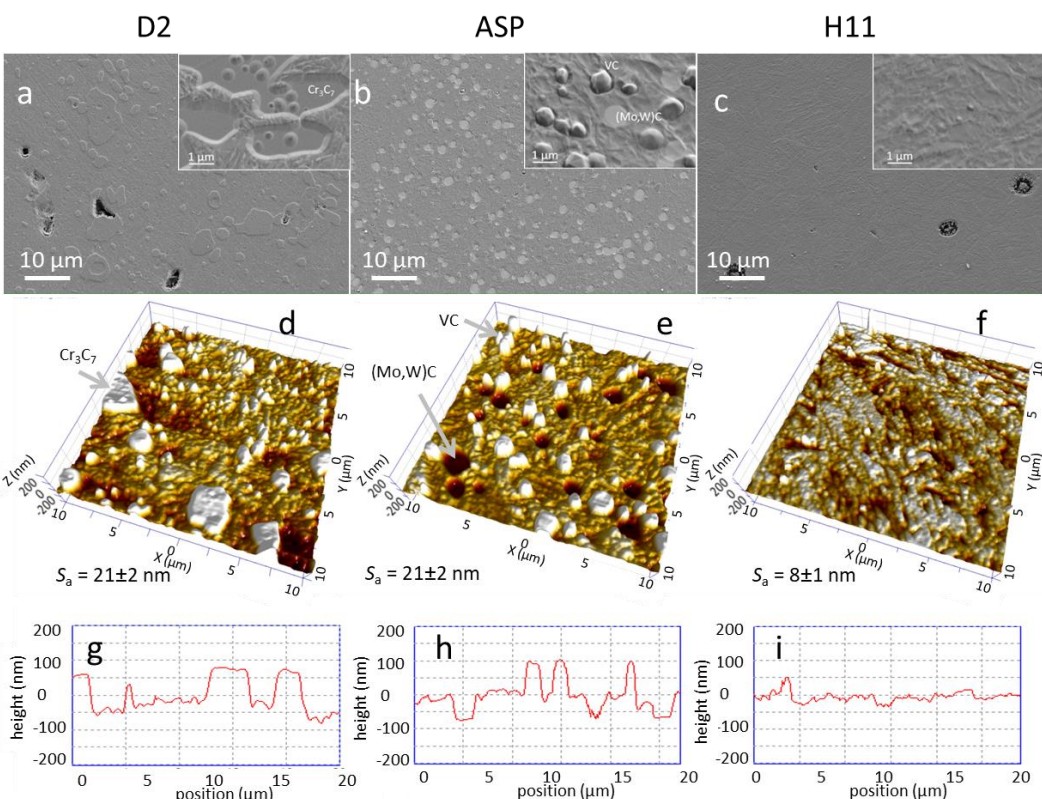

**Figure 5.** SEM images (**a**–**c**), AFM images (**d**–**f**), line profiles (**g**–**i**) and surface roughness ($S_a$) of D2, ASP30 and H11 tool steel substrates after RF ion etching in CC7 deposition system. The scan area and z-scale are 20 μm × 20 μm and 200 nm, respectively.

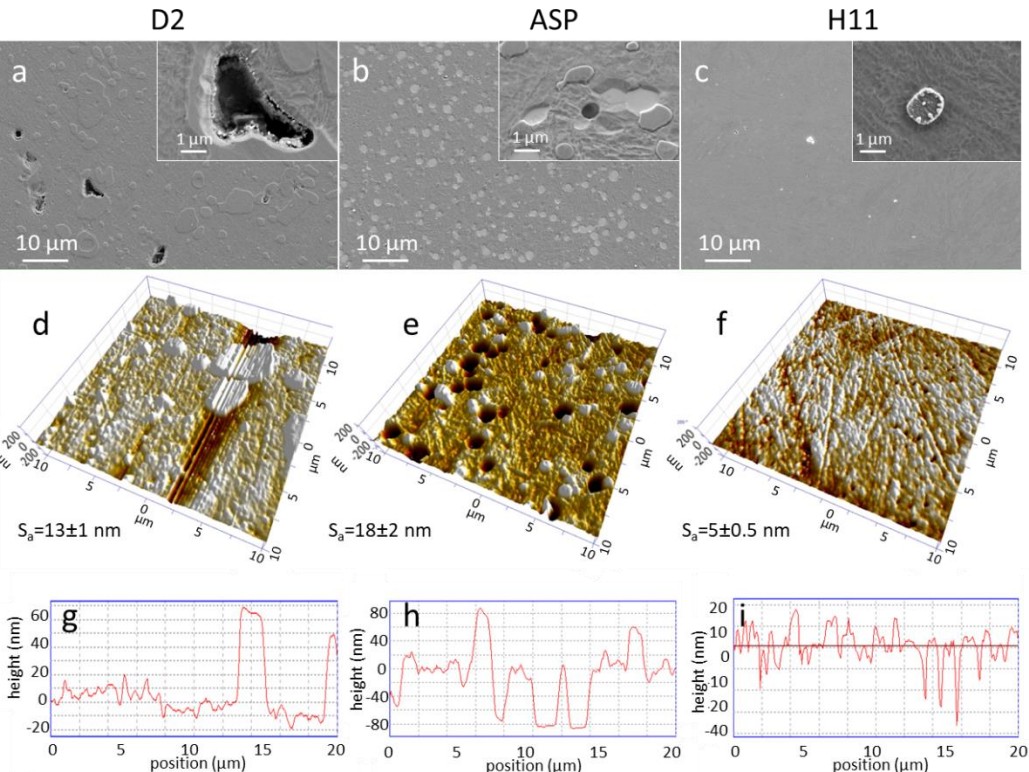

**Figure 6.** SEM images (**a**–**c**), AFM images (**d**–**f**), line profiles (**g**–**i**) and surface roughness ($S_a$) of D2, ASP30 and H11 tool steel substrates after DC ion etching in BAI deposition system. The scan area and z-scale are 20 μm × 20 μm and 200 nm, respectively.

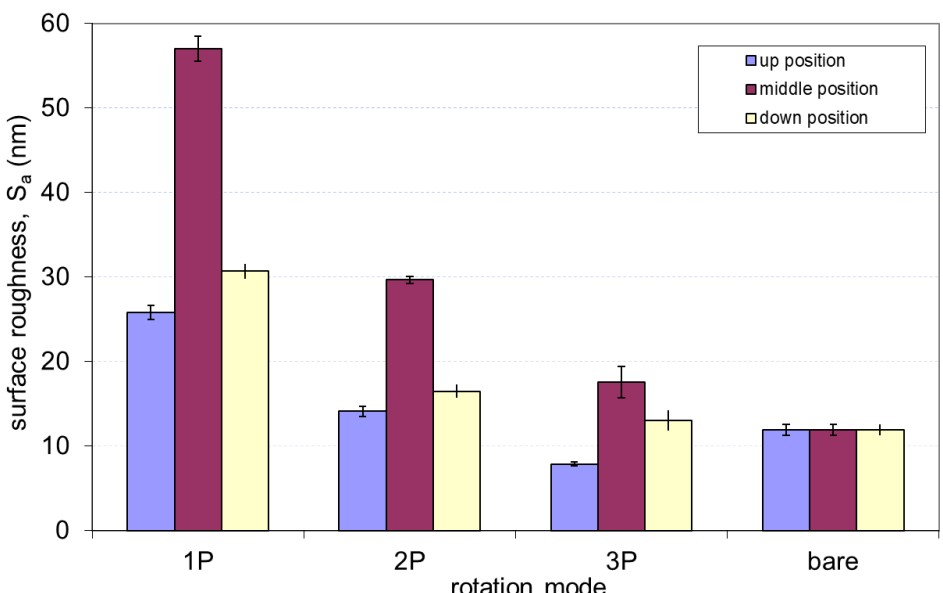

**Figure 7.** The 3D surface roughness ($S_a$) after filament-assisted pulse etching in AIP deposition system. The etching procedure was performed in the same batch using onefold, twofold and threefold rotation of ASP substrates, while their vertical position was also different. The scan area was 300 μm × 300 μm.

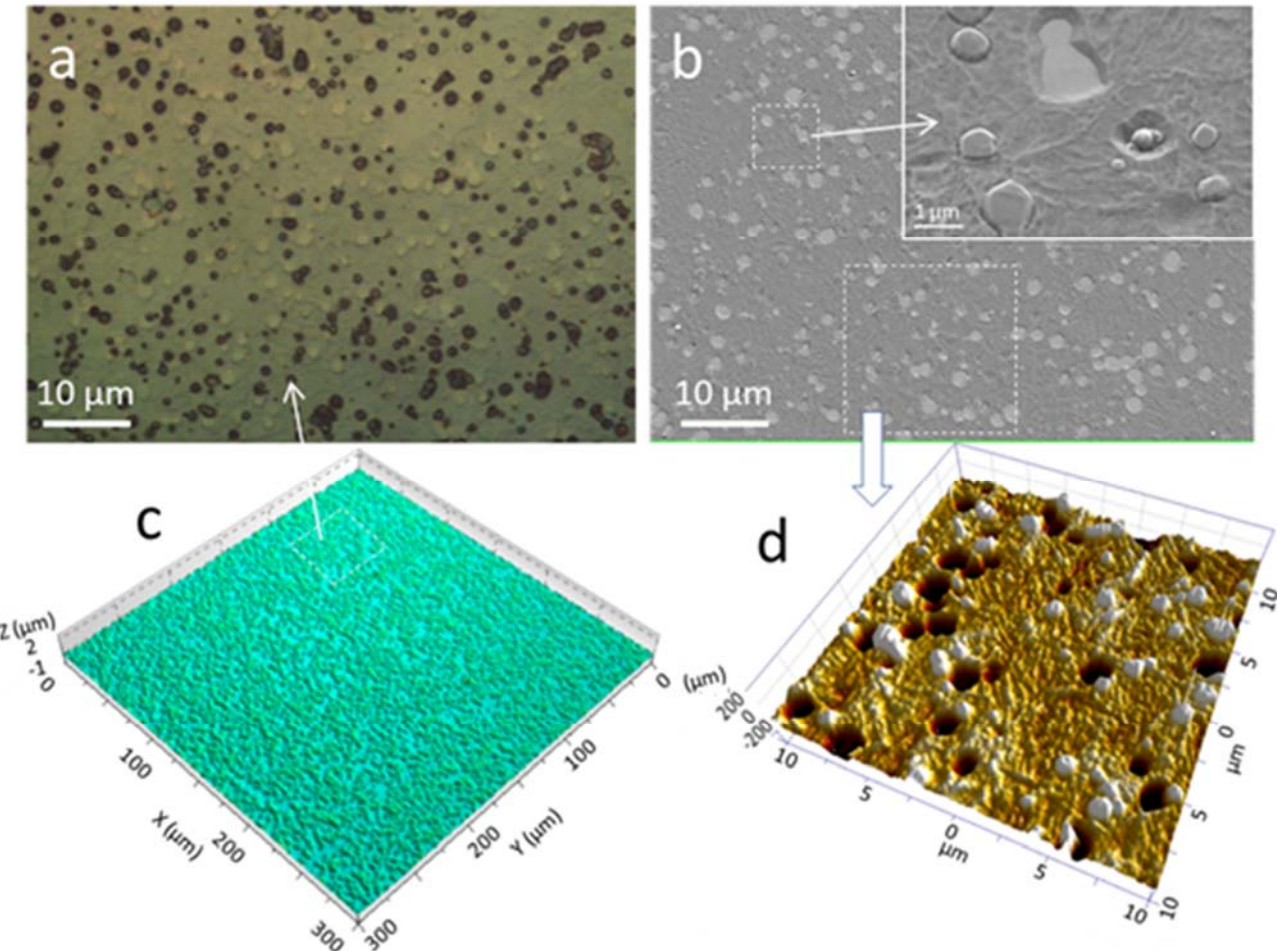

**Figure 8.** OM image (**a**); SEM images at lower (**b**) and higher magnification (see inset); 3D profile image (**c**); AFM image (**d**) of the ASP tool steel substrate surface after DC ion etching in BAI deposition system.

Etching of a surface by ion bombardment does not remove the contaminated surface layer without side effects. These side effects are implantation of bombarded ions, radiation damage, surface topography development, ion beam mixing, preferential sputtering and related changes in surface stoichiometry [24]. Furthermore, ion bombardment causes surface micro-roughening and the formation of various micro-sized surface features (Figure 9). Cones, pyramids, pits, hillocks, steps, etc., have been observed, and their formation is closely related to the initial surface irregularities, impurities, intrinsic or ion-beam-induced defects and variations in the sputtering yield as a function of the angle of ion beam incidence to the surface [25,26]. Especially large topographic changes occur at high fluencies of ions. Cones are the most common surface features produced by sputtering. They can be formed by different mechanisms. In 1971, Wehner and Hajicek [27] published a paper where they demonstrated that cones on etched substrates can arise as a result of a very small amount of certain foreign atoms (Figure 10), which are present as impurity atoms in the substrate material or are supplied during sputtering from another source. The islands of backscattered materials or contaminants act as local etch masks, as their sputtering rate is different in comparison with the surrounding material (matrix). This can cause the formation of various surface structures.

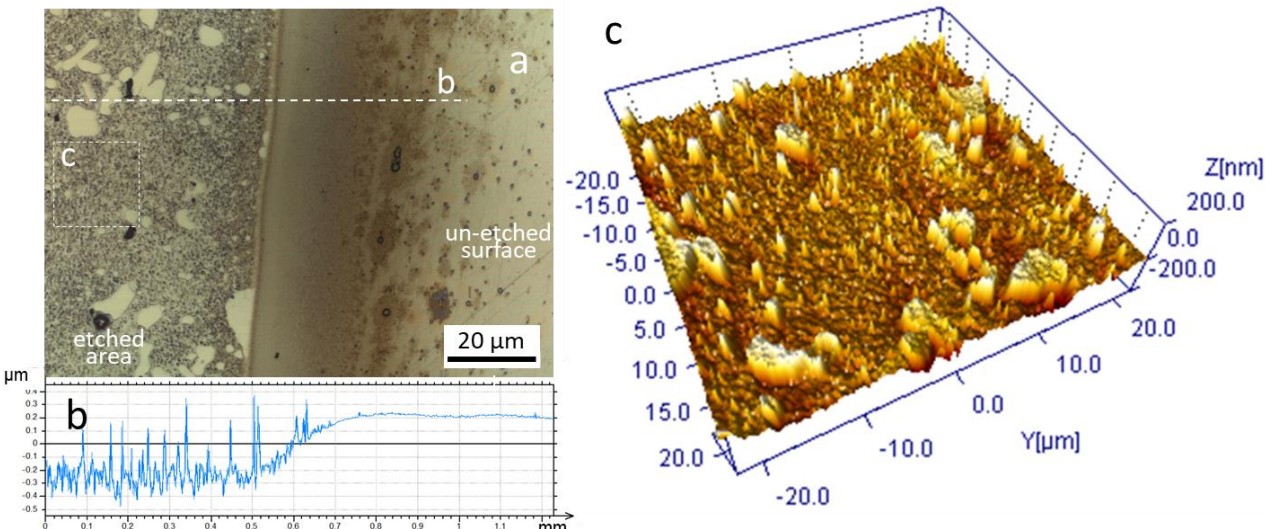

**Figure 9.** Optical image (**a**) and roughness profile measured along the dashed line (**b**) show the transition from the region exposed to ion etching to the unexposed one (this part of the substrate was covered with a stainless steel foil). Ion etching of D2 substrate was performed in the CC9 deposition system (MF + booster). The etching rate was quantified by measuring the step height between the etched and un-etched surface area (**b**). AFM image (**c**) of ion-etched area of D2 tool steel substrate (scan area was 50 μm × 50 μm).

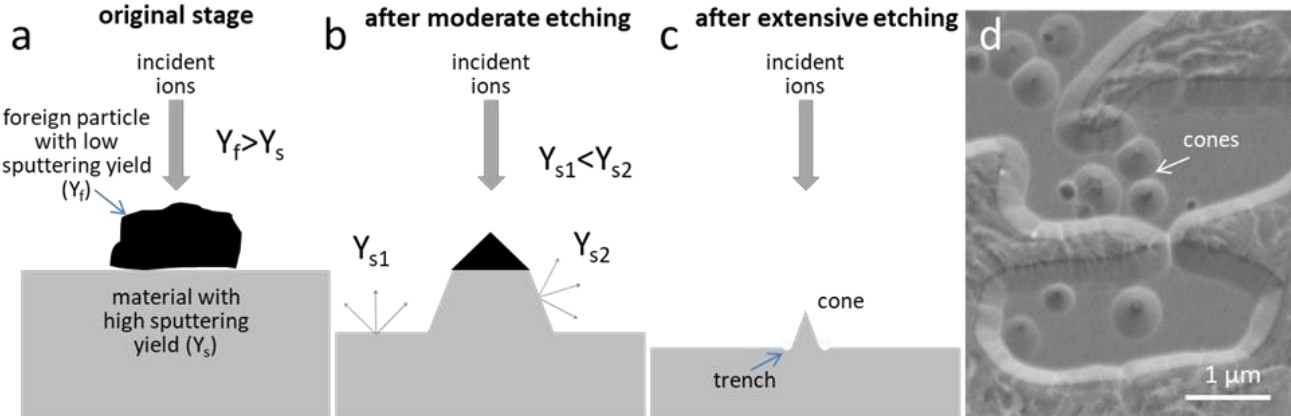

**Figure 10.** Scheme showing stages of cone formation during ion etching of substrate covered by a micrometer-sized foreign particle. If the foreign particle is located on the substrate surface, then it prevents the etching of the substrate area underneath the seed (**a**). Therefore, a step will be formed under the edge of the particle (**b**). As the particle itself shrinks, the step will be inclined. When the shielding particle is eroded away, a sharp cone of matrix material remains (**c**). Top view SEM image (**d**) shows the cones formed on the surface of carbides in D2 substrates during RF ion etching in the CC7 deposition system.

Cone-like features can also form underneath a foreign particle, which screens the underlying material from the ions. Figure 10 shows the scheme of the cone formation stages during ion etching of the substrate where a micrometer-sized foreign particle (with low sputtering yield) is present. As the sputtering proceeds, the area of this particle decreases, and the surrounding substrate material is sputtered away at a higher rate (Figure 10b). When the shielding particle is eroded away, a sharp cone of matrix material remains (Figure 10c). The size and shape of the cone depends on the ratio of particle shrinkage rate to the removal rate of bulk material. As the bombardment continues, the cone decreases in size, and it eventually disappears.

Cone-like structures can also form due to the initial micro-topography of the substrate surface (e.g., asperities). The variation in sputtering yield with the angle of ion incidence

causes the surface to erode more rapidly where the angle of incidence is higher. This is demonstrated in Figure 11, where we calculated sputtering yields as a function of $Ar^+$ incidence angle for several substrate materials investigated in this work. We used SRIM (Stopping and Range of Ions in Matter) software, which is based on the binary collision approximation to simulate the sputtering [28] process and calculate the sputtering yields and other sputtering parameters. The calculations were made for 5000 argon ions bombarding the surface with energy of 200 eV. It can be seen that sputtering yield strongly depends on the angle of ion incidence. In some cases, the sputtering yield at higher incidence angles can be twice or three times as high as the sputtering yield for perpendicular ion bombardment. Hence, the surfaces that are exposed to the ion flux under higher angles with respect to the surface normal will be etched much faster than those etched in the direction of the surface normal. These results in the preferential etching of inclined surfaces and promotes formation of faceted surfaces. It should be mentioned that the plasma sheath around the substrates is relatively large (around 1 mm) compared to the height of the protrusions (typically less than 0.5 μm). The plasma sheath, therefore, does not follow the topography of protrusions. This means that the ion flux at the protrusion arrives under various angles, while on the flat parts of the substrate (e.g., iron matrix of the steel), the ions bombard the surface predominantly in the perpendicular direction.

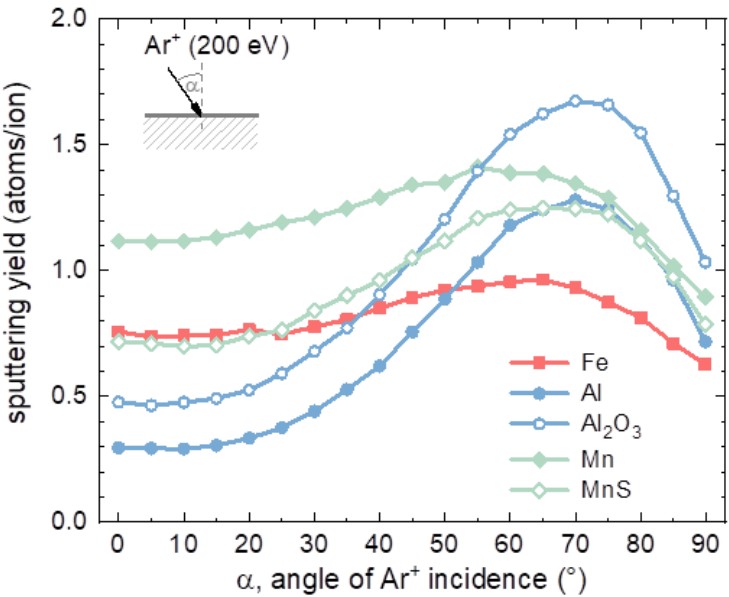

**Figure 11.** Sputtering yields as a function of $Ar^+$ incidence angle for several substrate materials investigated in this work. Calculations were performed by SRIM program.

The sputtering yield changes (decreases and also increases) when the surface of the substrates is faceted or roughened. Faceting usually starts at corners of protrusions (Figure 12), which always have some rounding, and therefore present a variety of incidence angles to the incoming ions (ranging from glancing to normal) [29]. This causes more rapid etching at the corner than on the flat surface, and simultaneously, the sputtering yield at the corner increases. Therefore, with increasing etching time, the corner is increasingly faceted. The angle of the faceted corner coincides with the angle at which the sputtering rate has the highest value. Thus, the angular dependence of the sputtering yield of the cone material largely determines the final shape of the cone. The enhanced etching rate at the sidewall causes the formation of cones at any protrusions on the substrate surface. The cones continue to sharpen as etching proceeds. Due to the higher sputtering (etching) rate at the sidewalls, the cones will eventually disappear after a longer period of etching. However, the substrate surface will at the same time expose new second-phase regions or impurity particles and new cones will start to form. Thus, a steady-state surface topography forms.

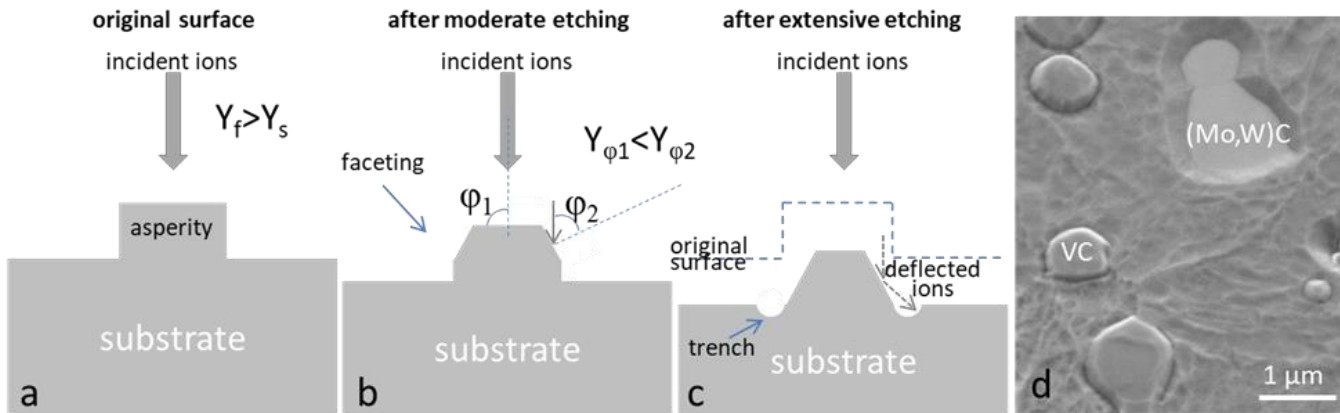

**Figure 12.** Schematic diagram outlining the mechanism of faceting and trench formation during ion etching of a protrusion on the substrate surface (**a**–**c**). Top view SEM image of ASP30 substrate after DC ion etching in BAI deposition system (**d**). In the SEM picture, we can see the faceting of carbides and the formation of trenches.

In general, the picture described above is more complex if other phenomena, such as surface diffusion of atoms on the substrate surface, are considered. Re-deposition and cross-contamination of sputtered materials, as well as sputtering by reflected ions, lead to additional changes in the sidewall shape. Cones are usually surrounded by shallow trenches which are created at the bottom of etched sidewalls (Figure 12c). Typically, trenches are formed due to enhanced erosion by incident ions reflected off the sidewall by energetic sputtered atoms from the cone sidewall. As a result, the cones develop grooves around them (Figure 12d).

The sputtering rate also depends on the texture of the coating or crystal orientation of individual grains and inclusions. Thus, when the polycrystalline substrate is ion etched, there will be a preferential sputtering due to the presence of individual grains and therefore, a stepped surface topography usually develops.

If the substrate material is composed of different phases of material, shallow craters and protrusions are generated during ion etching due to the difference in the sputtering rate between the different phases and matrix. For example, in ASP30 tool steel substrate, the sputtering rate of the $M_6C$ and the MC carbides is higher and lower than the martensitic matrix, respectively [30]. Thus, the consequence of ion etching is the formation of a large number of pits ((Mo,W)C carbides) and hillocks (VC carbides) that are evenly distributed over the substrate surface, as seen in Figure 13. Shallow craters and protrusions also form at sites of nonmetallic inclusions that have a higher (e.g., MnS) or lower (e.g., oxides) etching rate than the martensitic matrix (Figure 14). According to SRIM calculations shown in Figure 11, the sputtering yield of $Al_2O_3$ is lower than the sputtering yield of Fe for $Ar^+$ incidence angles below 40° but increases dramatically at higher angles. The sputtering yield of MnS is similar to the sputtering yield of Fe for lower incidence angles but increases substantially at angles above 30°. Hence, predicting the final surface topography of a material such as tool steel, which has many nonmetallic inclusions, is a difficult task since it depends on many local parameters as well as on the specifics of the etching procedure and substrate rotation. Nevertheless, our observations show that substrate roughness increases with the etching time.

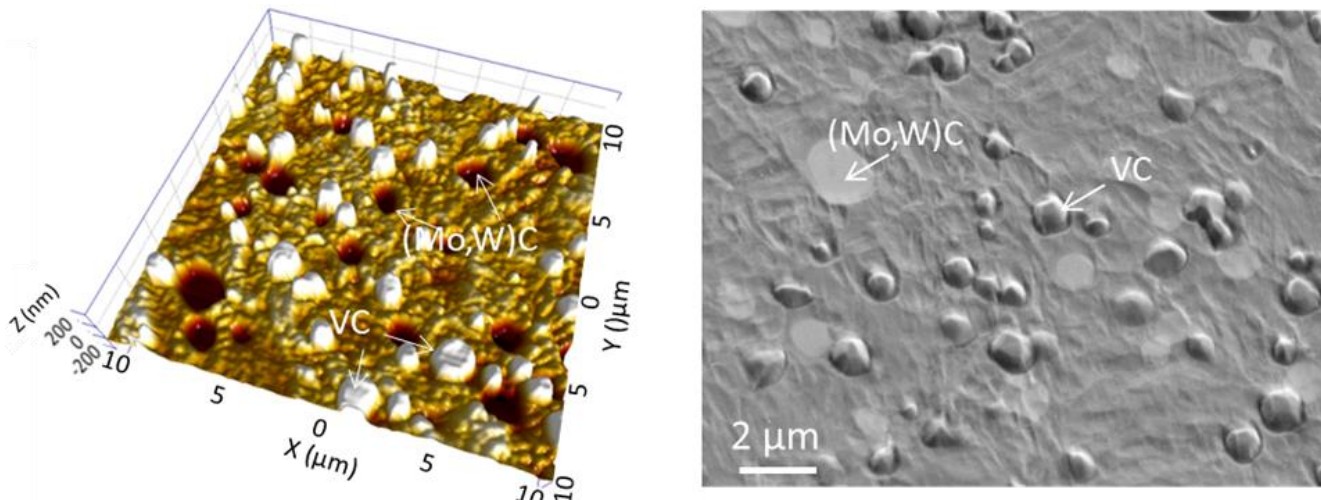

**Figure 13.** AFM and SEM images of the shallow protrusions and shallow craters formed by the RF ion etching of ASP30 tool steel substrate in the CC7 deposition system. The SEM image is a side view from about 45° inclined direction. Carbide inclusions do not retain their original profiles; they develop facets on the sidewalls and trenches at the bottom. The VC carbides become more protruding from the matrix and sharper after longer etching times.

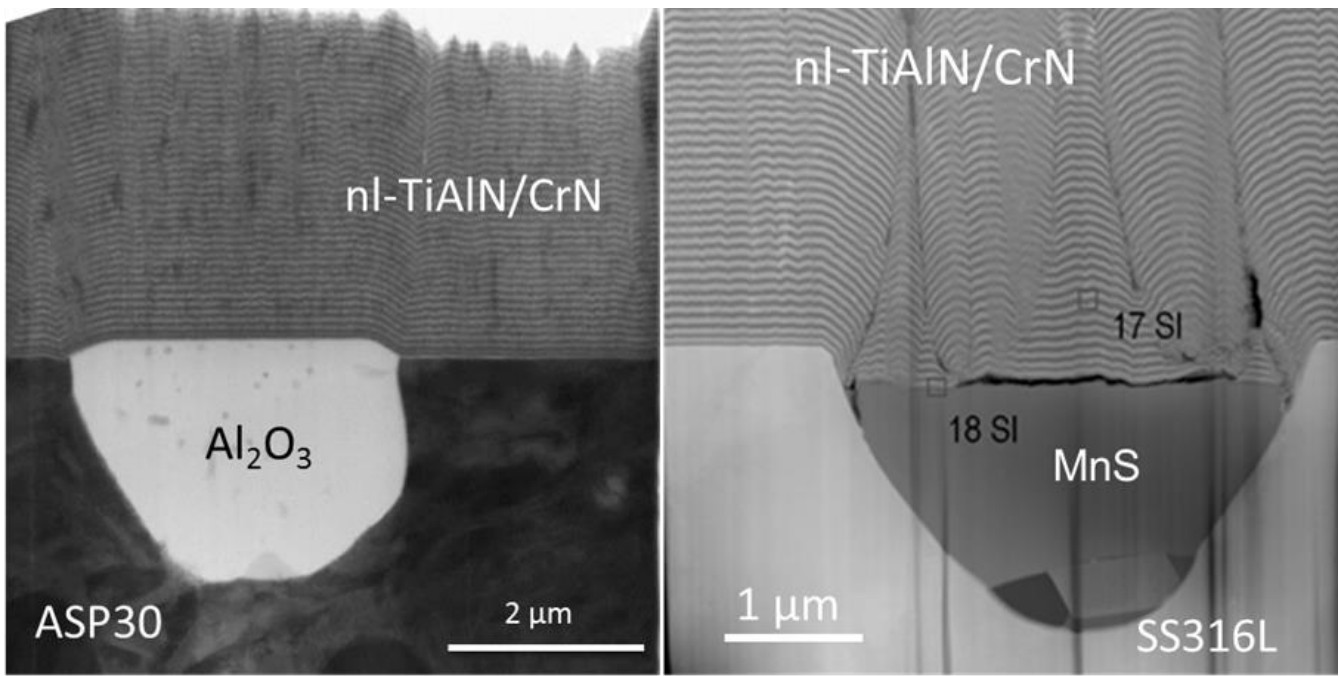

**Figure 14.** STEM image of FIB cross-section of the nanolayered TiAlN/CrN hard coating at sites of $Al_2O_3$ (ASP30 substrate) and MnS (SS316L substrate) nonmetallic inclusions.

As mentioned above, the etching efficiency depends not only on batching material and etching conditions, but also on the geometry of the substrates (tools), their positions in the vacuum chamber and the rotation mode [31,32]. On the industrial scale, PVD coatings are commonly deposited on substrates with complex three-dimensional shapes that have sharp edges (e.g., cutting tools), holes and slots. Ion etching on such substrates is nonhomogeneous due to varying ion currents arriving at the substrates. During a given deposition run as well as from one run to another, the etching rate is not only spatially dependent due to the different batching configurations but also temporally dependent due to multiple rotations of substrates. The effects of sample shape, orientation and distance have only scarcely been studied. Most of the available literature refers only to the etching

of flat substrates. Therefore, it is very important to study more in detail the differences in etching between 3D substrates and flat substrates that are usually used in tests.

A typical job-coating batch is filled with tools of different geometry, size and height. We can expect a highly non-uniform electric field distribution, not only around the different tools positioned at various heights within the batch but also on different parts of the same tool. For this reason, it is very difficult to predict the degree of etching on different parts of the tool. The knowledge on the etching efficiency within a batch therefore needs to be acquired empirically. An example of differences in the surface topography for the similar etching and coating conditions is shown in Figure 15. The figure shows 3D surface topography images of two D2 tool steel substrates after TiN deposition in the BAI deposition system. The individual samples were prepared in two different batches for the same etching and deposition parameters. The only difference between both samples was the batching configuration.

Since all topographic irregularities formed during ion etching are transferred to the coating surface, they can also be observed on the coated tools under an optical microscope. Under the microscope, the contours of carbides are clearly visible and more or less pronounced, depending on the etching efficiency (Figure 15a,b). It is evident that the TiN at the sites of carbide grains protrudes out of the surface much more on the sample in Figure 15a than on the sample in Figure 15b.

The etching efficiency can be quantified from roughness profiles (Figure 15c,d). The highest peaks in the 3D surface profile image (Figure 15e,f), which belong to the nodular defects formed during deposition, must be eliminated in the analysis. The height of steps at the sites of carbides is the measure of etching rate. Thus, for two different batching configurations, presented in Figure 15, the step heights are 165 nm and 94 nm, respectively. From the photos of both batches, we can conclude that the first sample was less shaded by other tools, so the ion flux density and consequently the etching rate was higher. Based on these and other similar observations, we conclude that the degree of etching is highly dependent on the batching configuration.

On tools with sharp edges and corners, the applied negative bias leads to non-uniform electric fields and non-uniform plasma densities over the different faces of the substrate surface and thus to the non-uniform bombardment of overall substrate surface. The current density and consequently the erosion rate are higher at the corners than on the flat surface. On the other hand, ion etching of holes and slits is lower due to shading effects and possibly lower electric field (Figure 16) [33]. This non-uniform ion bombardment leads to variations in surface topography over the substrate surface. Therefore, a major problem in using ion etching for substrate cleaning is how to obtain a uniform and controlled ion etching over a surface. The uniformity of the electric field is further worsened by the fixturing system. As a general rule, the best plasma system design is the one that is geometrically symmetric, which is, however, difficult to achieve in many cases. Thus, for example, in the case of a magnetron deposition system, the magnetic field confines electrons and increases the local plasma density in one region, which leads to a decrease in plasma density in another region [34]. In order to achieve a more uniform etching, multiple rotations of substrates are necessary. However, even multiple rotations of the substrates do not completely solve the problem of uniform etching. For example, an end mill can be etched quite uniformly around the main flank, but in the flute (on the rake face), the etching efficiency will be lower. The disadvantage of complex movement is that the etching efficiency of the substrate decreases with the increased degree of rotation.

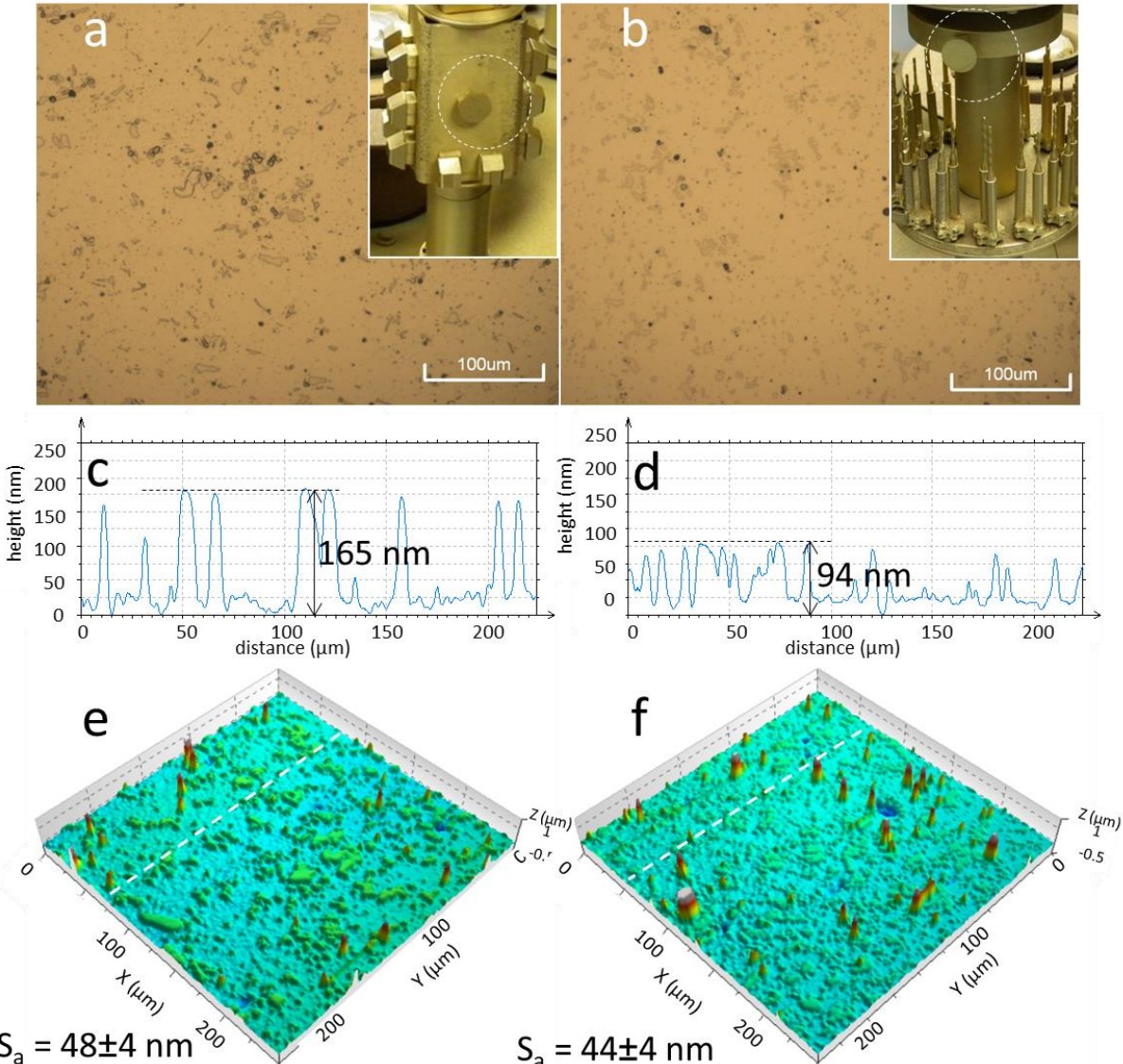

**Figure 15.** (**a**,**b**) OM images, (**e**,**f**) 3D surface profile images and (**c**,**d**) line profiles (along dashed lines in the Figure (**e**,**f**)) of TiN hard coating deposited on D2 substrate in the BAI deposition system at the same etching and deposition parameters. It is evident that the D2 substrate on the left was exposed to much more intensive ion etching (the step height at the carbide grain is 165 nm) than that right one (the step height at the carbide grain is 94 nm). The reason is the difference in batching configuration (see insets). The surface roughness, however, differs only slightly, due to the higher growth defect density on the right sample (406 defects/mm$^2$) in comparison with the left one (236 defects/mm$^2$). The nodular defects are the sharp peaks.

The erosion rate is higher at the corner than on the flat surface of the sample. This is attributed to the concentration of the electrical field lines around the corners and consequently, a higher ion current and erosion rate (Figure 16). Thus, we noticed that in the case of threefold rotation of the round plate substrate, the etching of the plate edge is much more pronounced than on the flat surface. This effect is much less pronounced for twofold rotation and the least for onefold rotation, where the electrical field lines are perpendicular to the substrate surface all the time. In a certain phase of threefold sample rotation, however, the sharp edge is highly exposed to intense ion bombardment. For example, Figure 17 shows the SEM images of D2 round substrates (5 mm in thickness, 16 mm in diameter) coated in the CC9 deposition system with nanostructured hard coating using threefold rotation. The ion etching is much more pronounced at the edge of the round sample ($S_a$ = 195 nm) than on the flat surface in the middle ($S_a$ = 54 nm) of the sample.

Due to variation in the electrical field at the substrate surface, the distribution of ion current density during etching and deposition is highly non-uniform, and at sharp edges, it may be many times higher than in the middle. Therefore, we can expect different coating properties and performance compared to the coating on the flat part of the sample. The majority of scientific papers concentrate on ion etching and subsequent coating deposition on a flat substrate. Only a few studies reported the differences between coatings deposited on the curved and flat substrates [15,35]. However, care must be taken when applying the observations made on the flat test samples to the tools with complex geometry.

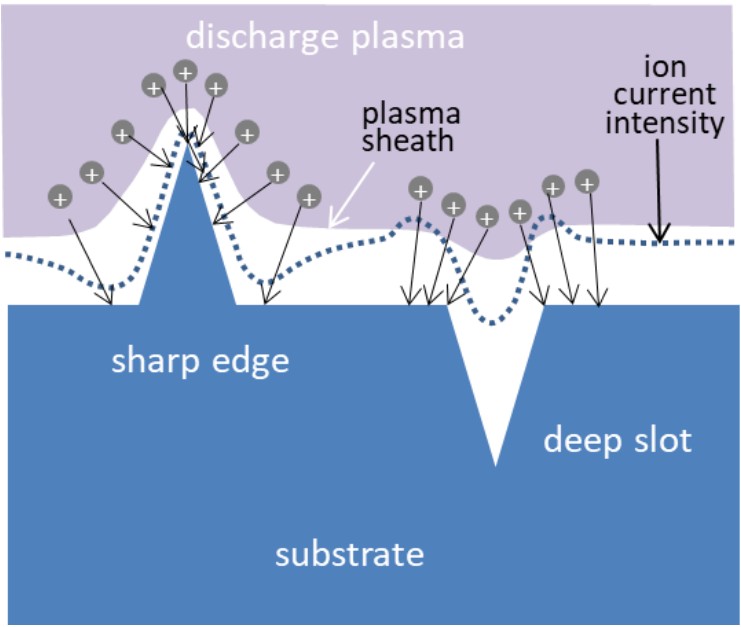

**Figure 16.** The sheath between homogeneous plasma and negatively biased substrate with complex geometric shape (sharp edges, slots) and the corresponding ion current density.

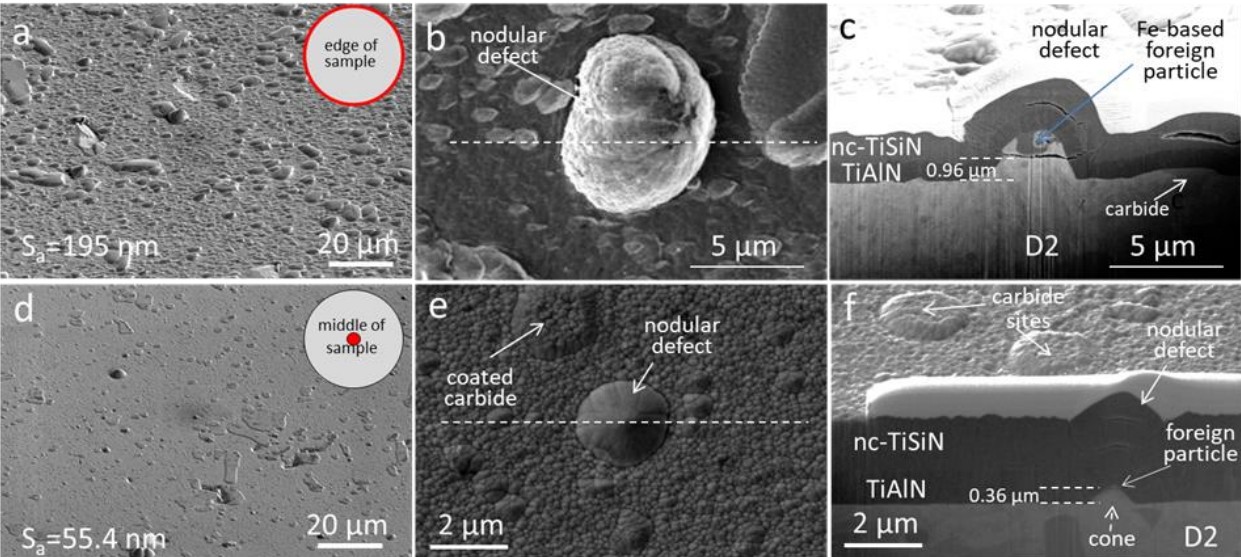

**Figure 17.** SEM images of the nanostructured hard coating TiAlN/TiSiN/TiAlSiN deposited in the CC9 deposition system: (**a**–**c**) at the edge and in (**d**–**f**) the middle of D2 round substrate. The SEM images (**a**,**d**) are the side view from about 45° inclined direction. Top view SEM images in the middle show selected nodular defects (**b**,**e**) and their FIB cross section (**c**,**f**). A step formed under the seed particles (marked with dashed lines) shows that a 3-times thicker layer was removed on the edge during ion etching in comparison with the middle of the substrate.

### 3.2. Micro Topography of Coating Surfaces

As mentioned above, all topographical irregularities on the substrate surface formed during pretreatment are transferred through the coating. Therefore, the intensity of ion etching can also be estimated after coating deposition, either using optical, AFM and SEM microscopes or a 3D profilometer. However, after coating deposition, additional topographic changes occur related to the intrinsic micro- and nanomorphology of the coating itself and growth defects (Figure 18). Low magnification observations (Figure 18a) show a smooth coating surface interrupted by unevenly distributed growth defects (nodular defects, pinholes, craters) of various shapes and sizes (see inset b in Figure 18a). At a high magnification of the area between growth defects, columns with dome-shaped tops can be observed. At even higher magnifications, the nanoscale sub-cells on the top of every columnar grain are visible (see inset d in Figure 18c).

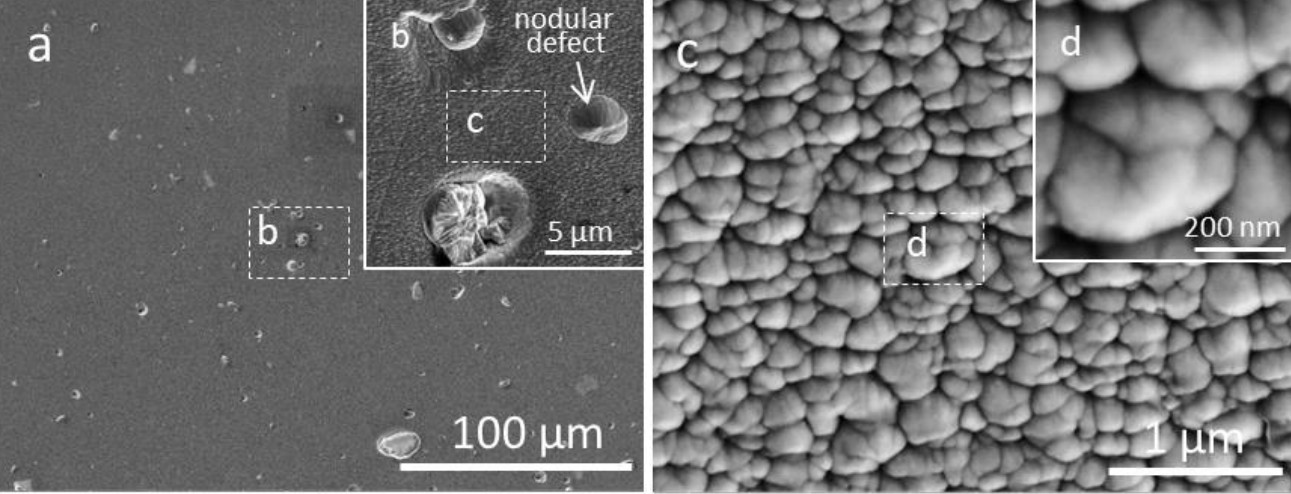

**Figure 18.** SEM observations of (**a**) macro-, (**c**) micro- and (**d**) nano-morphology of a magnetron sputtered nanolayer and nanocomposite coating TiAlN/TiSiN/TiAlSiN deposited on D2 tool steel substrate. Inset (**b**) shows the typical growth defects formed during deposition process.

The micro morphology of the depositing film is determined by the surface roughness and the surface mobility of the depositing atoms. If the substrate surface is rough, the variation in the angle of incidence causes the formation of a less dense coating with a more complex morphology. The peaks of protrusions receive the flux of depositing atoms from all directions and, if their surface mobility is low, the peaks grow faster than the valleys. The self-shadowing effect is even greater if the flux of depositing atoms is off-normal because the valleys are in "deeper shadows" than when the flux is normal to the surface [36]. The consequence of such film growth is the formation of a columnar microstructure with dome-shaped surface topography (Figures 19 and 20). The roughness of interfaces that are rather smooth near the substrate gradually increases towards the top of the coating (Figure 20). Rounded columns result in higher cumulative surface roughness of the coating, which increases with the coating thickness. The columnar morphology forms regardless if the material is crystalline or amorphous [37]. The driving force for columnar grain growth is surface energy minimization. Additionally, the self-shadowing effect increases the degree of roughening and causes a voided columnar structure.

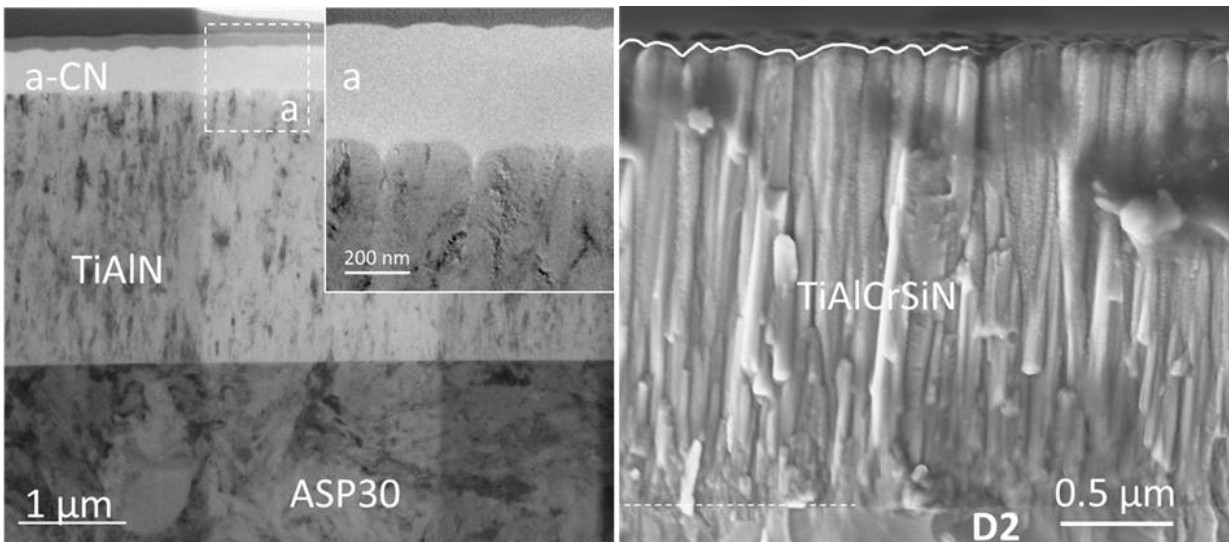

**Figure 19.** TEM cross-sectional image of a-CN/TiAlN and fracture cross-section SEM image of a TiAlCrSiN hard coating show how columnar microstructure changes the surface topography. It is evident that the surface roughness of the coating has its origin in the columnar microstructure. Both coatings were sputter deposited in the CC7 deposition system on tool steel substrates (ASP30, D2).

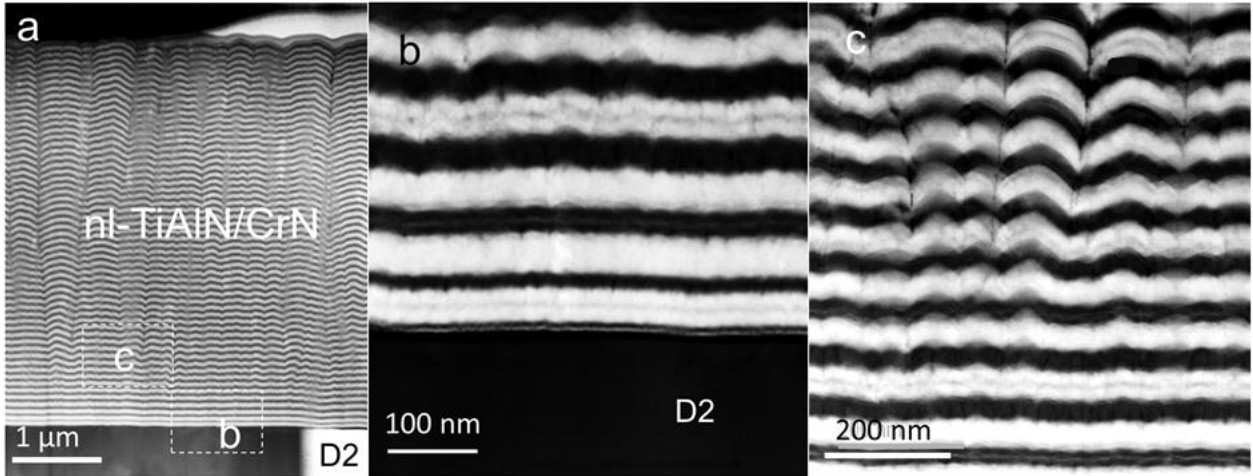

**Figure 20.** Bright-field STEM cross-sectional images of TiAlN/CrN nanolayered coating sputter deposited in CC7 deposited on D2 tool steel substrate (**a**). The roughness of the TiAlN/CrN interfaces gradually increases up to the top of the coating (**b**,**c**).

The surface roughness can be reduced by using high mobility depositing atoms. Their mobility depends on their energy, substrate temperature and interactions with the substrate surface (i.e., chemical bonding). For example, the most common way to increase the surface mobility of atoms is by low-energy ion bombardment during the deposition.

This simultaneously promotes chemical reactions and introduces new nucleation sites. The columnar growth can be disrupted by intense ion bombardment of the growing coating. For example, transition metal–nitride coatings produced by magnetron sputtering typically exhibit a pronounced columnar microstructure (Figures 19b and 21). However, if the deposition is interrupted and the coating is then exposed to ion bombardment for a certain time, new nucleation sites for the subsequently deposited nitride coating are created. Such intermediate ion etching causes an interruption of the coating columnar growth and eliminates the porosity along the columns. This is reflected in a fine-grained and less porous microstructure of the top layer. A change in the surface topography from dome-shaped column tops (Figure 21b) to a dimpled surface occurs (Figure 21c). After intermediate ion etching, the roughness of the coating surface decreases because

the sides of coating protrusions etch faster than the top. This consequently leads to the shrinking of the protrusions and even the elimination of some smaller cones [38].

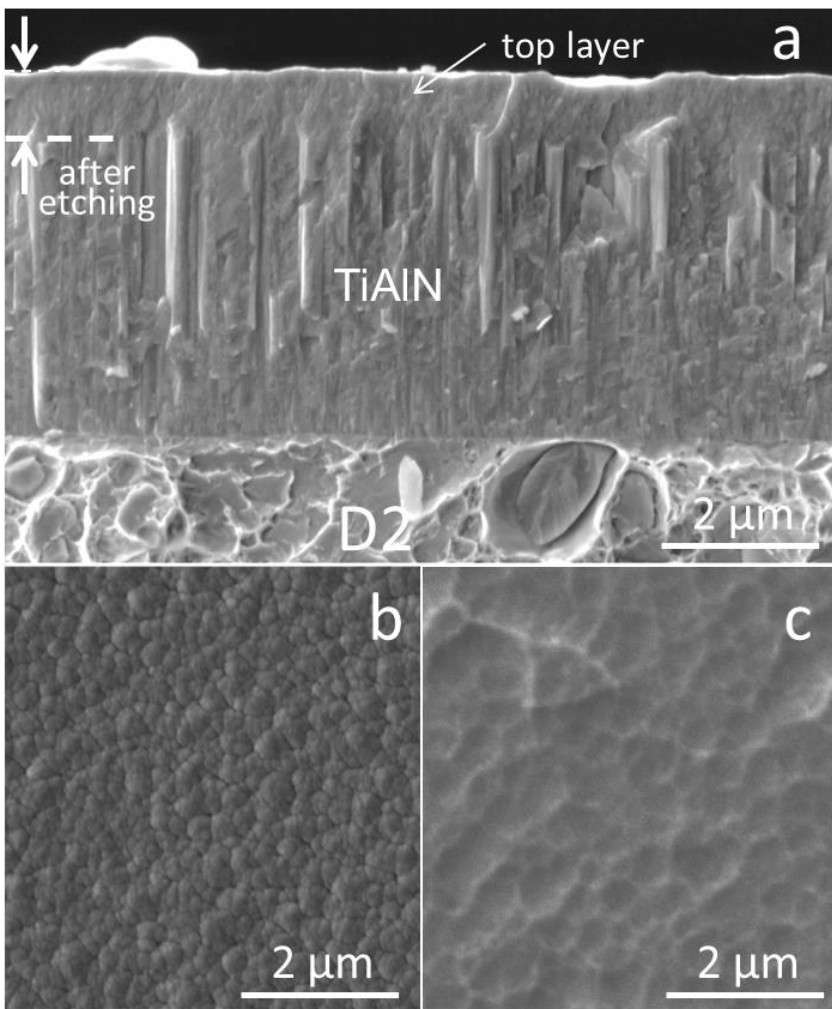

**Figure 21.** (**a**) Fracture cross-sectional SEM image of TiAlN coating sputter deposited in CC7 on D2 tool steel substrate; (**b**) top-view SEM image of the TiAlN coating prepared without (**b**) and with (**c**) an intermediate ion etching step.

The effect of ion bombardment during deposition strongly depends on the ratio between the deposition rate and the ion current density to the substrate, which should be as low as possible. For the selected deposition method, this ratio is influenced by the target-to-substrate distance, the substrate geometry, substrate rotation mode and batching configuration. At a certain bias voltage on the substrates, this ratio depends mostly on the rotation mode and is the highest for onefold rotation. We also have to consider that both the angle-of-incidence of the depositing atom flux and the deposition rate change constantly during the deposition process. Therefore, it is practically impossible to obtain a uniform and controlled ion bombardment over the surface of the complete batching material. The use of a substrate bias, however, leads to problems when depositions are made onto complex and irregularly shaped tools, with sharp edges and corners. In this case, an increased ion current density around the edges causes either an increased rate of re-sputtering of the deposited material [39] (e.g., magnetron sputtering) or a thickness increase (cathodic arc evaporation) [40]. This effect can be eliminated by proper selection of deposition parameters (e.g., bias voltage, deposition rate, batching configuration, the target-to-substrate distance).

Hard coatings prepared by various deposition techniques and deposition parameters exhibit a wide variety of microstructures which are reflected in different sizes of grains,

textures and morphology. For example, Figure 22 shows three different coatings prepared by different deposition methods on the same type of substrate material (ASP30). There is a significant difference in surface morphology between the different coatings. It seems that the initial nucleation of the coating follows the carbide structure in the steel substrates. Thus, on the TiAlN/Al$_2$O$_3$ coating sputter deposited in CC9, small pits appear at sites of (Mo,W)C carbides in the ASP30 tool steel substrate (Figure 22a). The top Al$_2$O$_3$ layer in this coating was deposited by the pulsed magnetron sputtering technique. Very small protrusions at sites of VC carbides can also be observed. On the other hand, pronounced protrusions are formed on the nl-TiAlN/TiN coating prepared in the CC9 deposition system at sites of VC carbides (Figure 22b). The surface topography is completely different when a TiN coating is prepared on ASP30 substrate in the BAI deposition system. It is characterized by dimpled surface topography (Figure 22c). The fracture cross section SEM images also reveal that at the sites of selected VC carbides, epitaxial growth of TiN coating occurs (see inset in Figure 22b,c). This phenomenon, however, does not occur at (Mo,W)C carbide sites and martensitic matrix, where a dense columnar morphology can be observed. All these phenomena show a decisive impact of the microstructure and the micro topography of the substrate surface, formed during ion etching, on the coating topography.

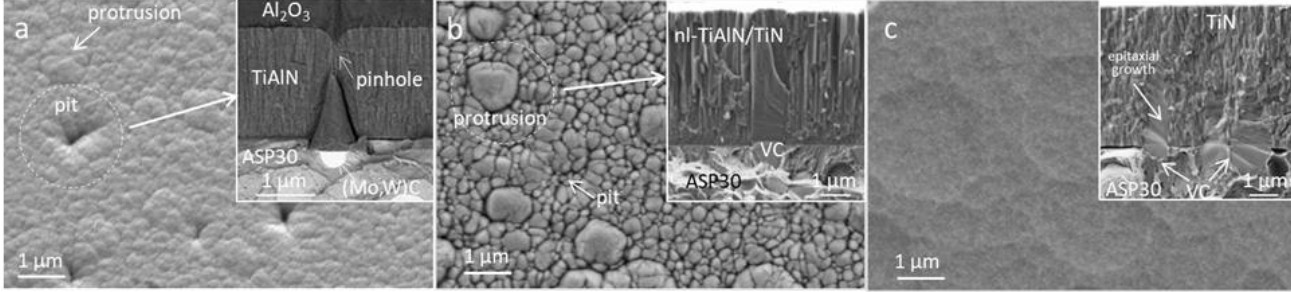

**Figure 22.** Top view SEM images of TiAlN/Al$_2$O$_3$ (**a**), nanolayered TiAlN/TiN (**b**) and TiN (**c**) hard coatings deposited on ASP30 tool steel substrates in CC7 (**a**,**b**) and BAI deposition systems. Fracture cross sections of the individual coatings are shown in the insets.

SEM images (Figure 23) present the observations of macro- (a–c) and micromorphology (d,e) of TiN hard coatings deposited on D2 tool steel substrates by three different deposition methods: low-voltage arc evaporation (a,d), magnetron sputtering (b,c) and cathodic arc evaporation (c,f). There are obvious differences in coating topography. Differences in topography are not only the result of different ion etching modes, but they also depend on the ratio of the deposition rate and the ion current density to the substrates, which is specific for the selected deposition method.

In the first of these two experiments, we analyzed the morphology of different hard coatings which were deposited by different methods on the same type of substrate (ASP). In the second one, we analyzed the TiN coating deposited on D2 substrates by three different methods. In the following, however, we will address the morphology of TiAlN hard coating, sputter-deposited in the same batch of the CC7 deposition system, on three different types of substrates. Coating topography was analyzed at macro and micro levels using 3D profilometry, AFM and SEM microscopes (Figure 24). At the micro-level (see SEM images), no significant difference in topography can be observed. For all three substrate materials, the dimpled surface topography of the TiAlN coating is characteristic. AFM images that were recorded at a scanning area of 20 μm × 20 μm, reveal a similar surface topography for all three tool steel substrates. Additionally, on coated ASP and D2 substrates, protrusions are visible at sites of carbides. The difference in surface roughness is not very large. It is the largest for D2 (S$_a$ = 30 nm) and the lowest for H11 (S$_a$ = 22 nm) substrates. On the other hand, the surface roughness determined by the 3D profilometer is much larger because the measurements that were performed at a scanning area of 300 μm × 300 μm, and also include the contribution of growth defects. The line profiles

were recorded on the defect-free areas. We can see that the surface roughness of TiAlN coatings deposited on ASP and D2 substrates is comparable and much higher than that deposited on the H11 substrate.

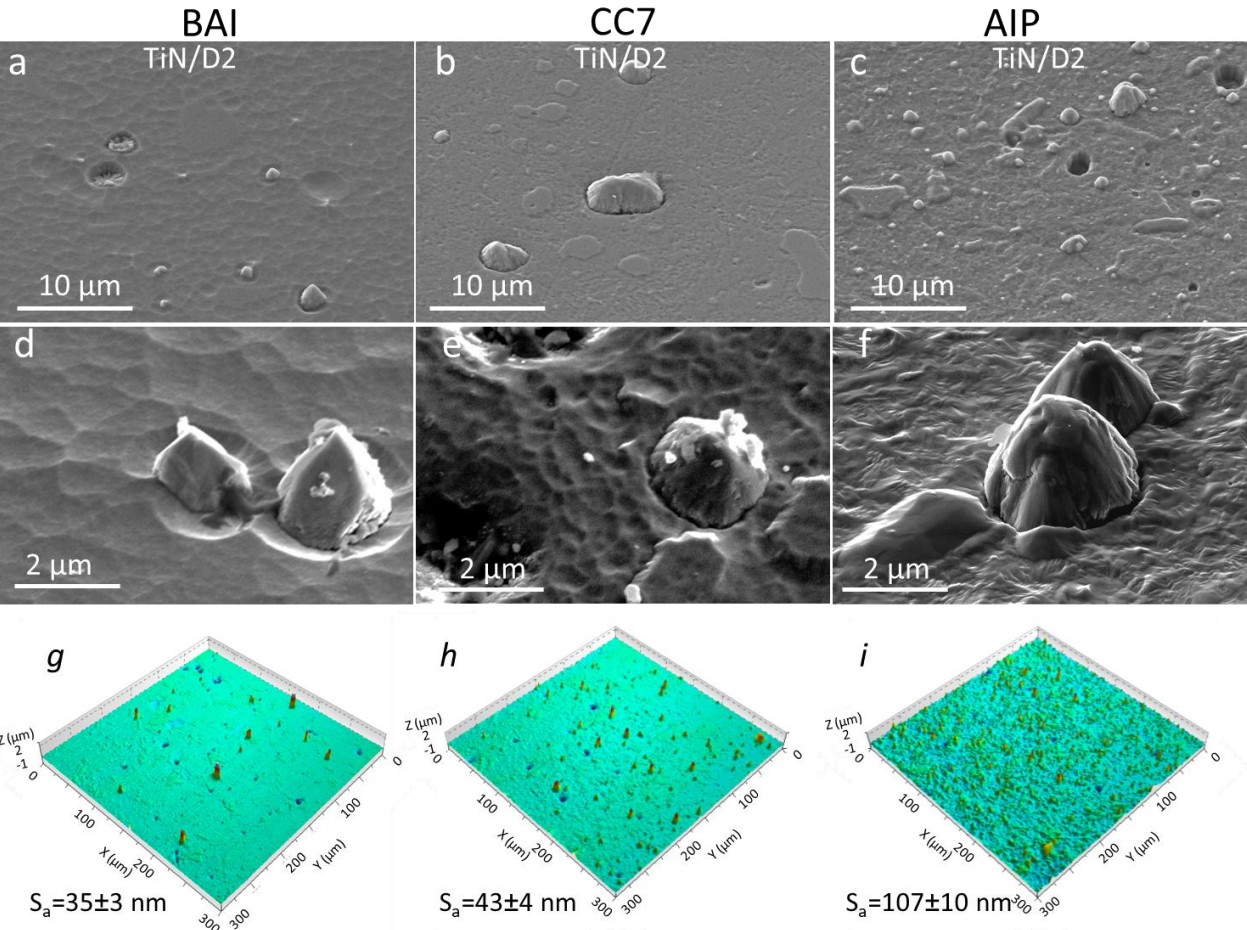

**Figure 23.** Top view SEM images at lower (**a**–**c**) and higher (**d**–**f**) magnifications and 3D surface profile images (**g**–**i**) of TiN coating deposited on D2 tool steel substrate using BAI, CC7 and AIP deposition systems. The SEM images are the side view from about 45° inclined direction. The sharp peaks in 3D surface profile images are the nodular defects while the blue dots are craters.

In the next test, the nl-TiAlN/TiN hard coating, prepared by magnetron sputtering in the CC9 deposition system, was applied on four different substrates (ASP30, D2, H11, SS 316L). One-third of the substrates were cleaned with MF ion etching and one-third with a combination of MF and "booster" ion etching, while one-third of the samples were not exposed to etching. SEM observations of surface micro topography, presented in Figure 25, show that there are only small differences in the intrinsic morphology of all coatings. All samples are characterized by a dome-shaped morphology of the coating surface in the area without growth defects. The average diameter of the columns is comparable for all samples, regardless of the type of substrate and the method of etching. Similar surface morphology of the coating also occurs on the surface of the carbides. This could mean that the geometrical shadowing effect plays a decisive role in the formation of a columnar microstructure. We assume that the etching of the martensitic matrix in individual steel was similar.

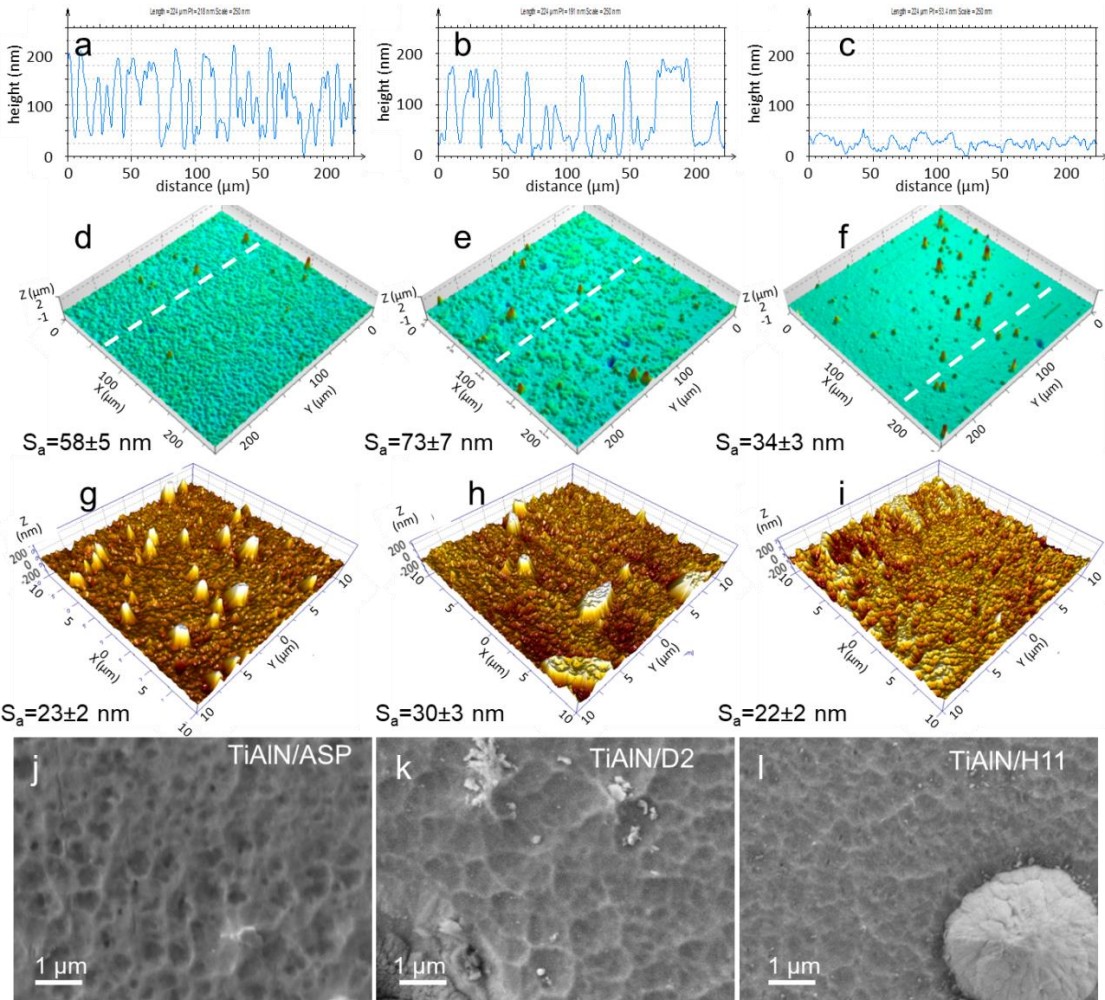

**Figure 24.** The 3D surface profile images (**d–f**), AFM images (**g–i**) and top-view and SEM (**j–l**) images of TiAlN hard coating sputter deposited in CC7 system on three different tool steel substrates (ASP30, D2, H11). The sharp peaks in 3D surface profile images are the nodular defects while the blue dots are craters. The line profiles (**a–c**) were recorded on the defect-free areas of 3D profile images (**d–f**).

In another test, we deposited different hard coatings on D2 tool steel substrates in the deposition system CC9. The top-view SEM images of different coatings are given in Figure 26. We can see that four different coatings have a similar micro- and nano-morphology, which is similar to cauliflower. This test indicates that the chemical composition of the coating has no effect on its morphology.

The analysis of microstructure and topography shows that there are considerable differences between the coatings deposited in the same batch but using different modes of substrate rotation (Figures 27 and 28). In the case of onefold rotation, the microstructure is characterized by a well-developed columnar microstructure with pores between the individual columns and rough topography. In two- and threefold rotations, the distance to the target and orientation changes significantly during the deposition process. The first consequence is that the incidence angle of the metallic atoms and ions changes all the time, which works strongly against the directional columnar growth. The second consequence arises when the sample is facing away from the target (this applies mainly for threefold rotation) but is still exposed to the bombardment of ions. Namely, the ions from bulk plasma follow the electric field, which is confined in the plasma sheath of the substrate (a few mm). If an ion arrives near the edge of the sheath, it is accelerated toward the substrate. This can also occur in the areas in shadow of the particle flux from the target. In the threefold and partly in the twofold rotations, the periods of growth are followed

by periods of densification, which ensures the formation of a compact, dense film [41]. The individual columns are much more pronounced in 1-fold rotation than 2-fold and 3-fold rotations. The microstructural differences between the samples prepared by two- and threefold rotations are relatively modest.

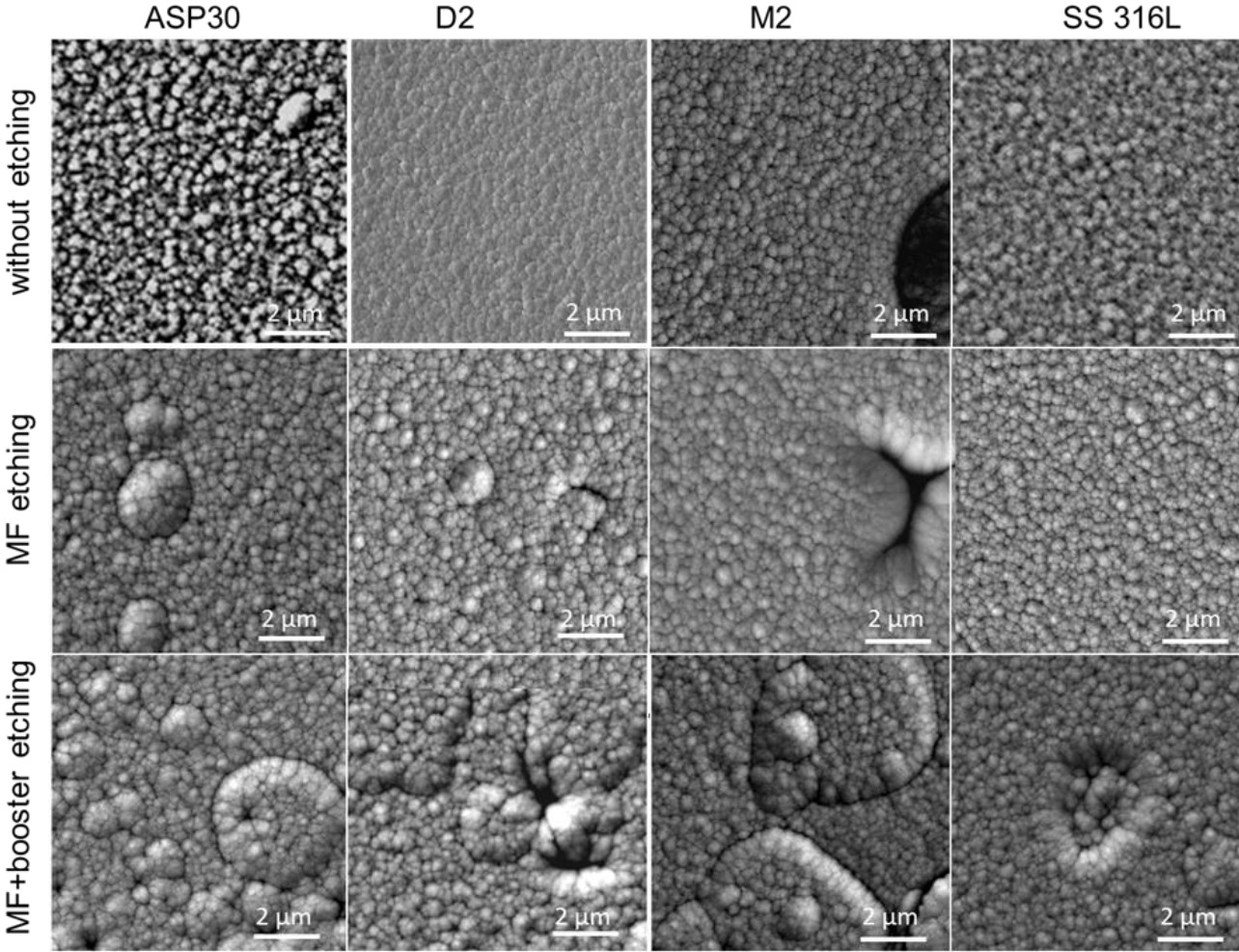

**Figure 25.** Top-view SEM images of surface topography of nanolayer TiAlN/CrN hard coatings deposited on un-treated, MF and MF + booster etching tool steel (PM ASP30, M2, D2) and stainless steel SS3016L substrates. There are no significant differences in topography.

The effect of ion bombardment on the morphology and topography strongly depends on the type of ion species bombarding the growing coating. There are two sources of ions: working gas ions (i.e., $Ar^+$ and $Kr^+$) or metal ions from the target material. In the low-voltage electron beam evaporation (BAI system) and cathodic arc evaporation (AIP system), a large share of the evaporated target material is ionized. This is not the case with the conventional magnetron sputtering process, where the ionization degree of the sputtered metal is typically low. Both sputter deposition systems used in this study (CC7, CC9) are equipped with a high ionization module (HIS), which can improve the ionization of the sputtered metal atoms to more than 50% [18]. The highest ionization degree and charge state of the target material is achieved in the cathodic arc processes (AIP). Differences in the method of evaporation of the target material, the degree of ionization, the type of ions, the ion flux density on the substrates and the deposition rate are also reflected in the microstructure and surface morphology of the coating prepared by different deposition methods. In general, the higher degree of ionization and higher energy of metal atoms means higher coating density.

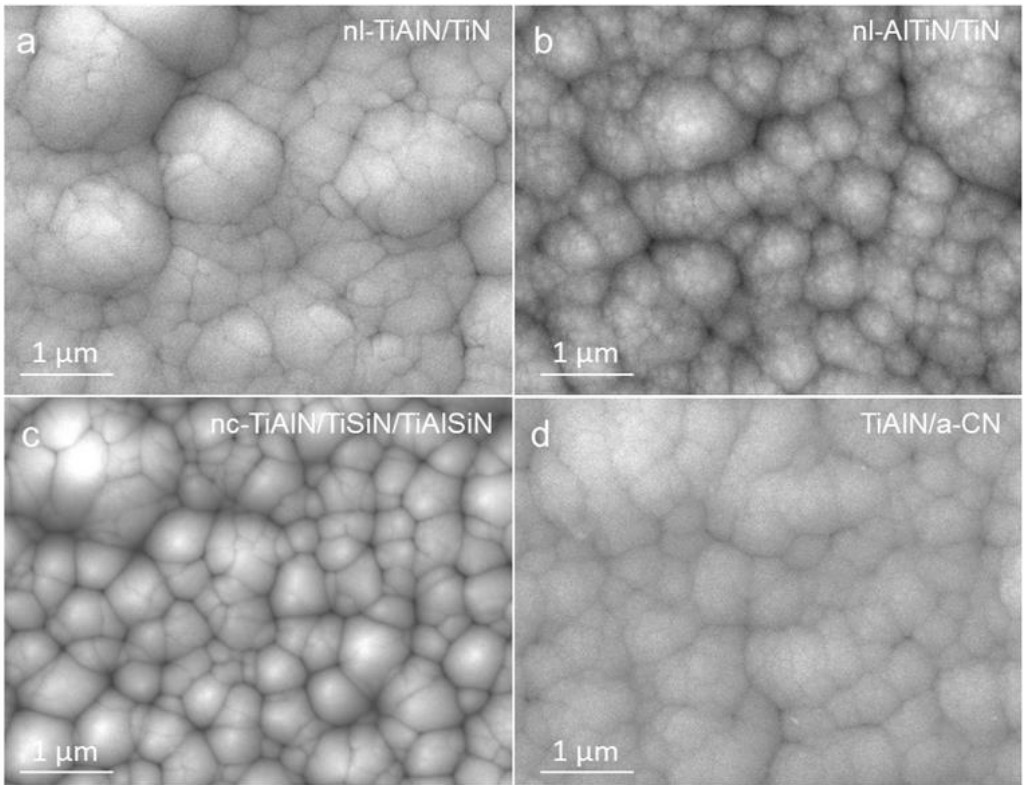

**Figure 26.** Surface morphology of four different hard coatings: nl-TiAlN/TiN (**a**), nl-AlTiN/TiN (**b**), ns-TiAlN/TiSiN/TiAlSiN (**c**) and TiAlN/a-CN (**d**). They were deposited on D2 tool steel substrates in the CC9 system using 1-fold rotation.

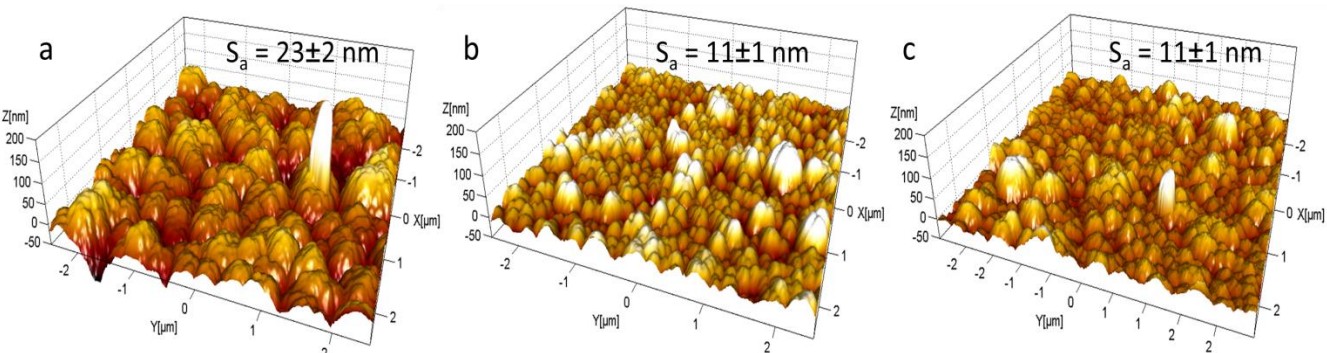

**Figure 27.** AFM images (scan area 5 μm × 5 μm) of double layer TiAlN/a-CN hard coating sputter deposited on D2 tool steel substrates in the CC9 system. The samples were prepared in the same batch but rotated around a different number of axes: (**a**) onefold, (**b**) twofold, (**c**) threefold rotation.

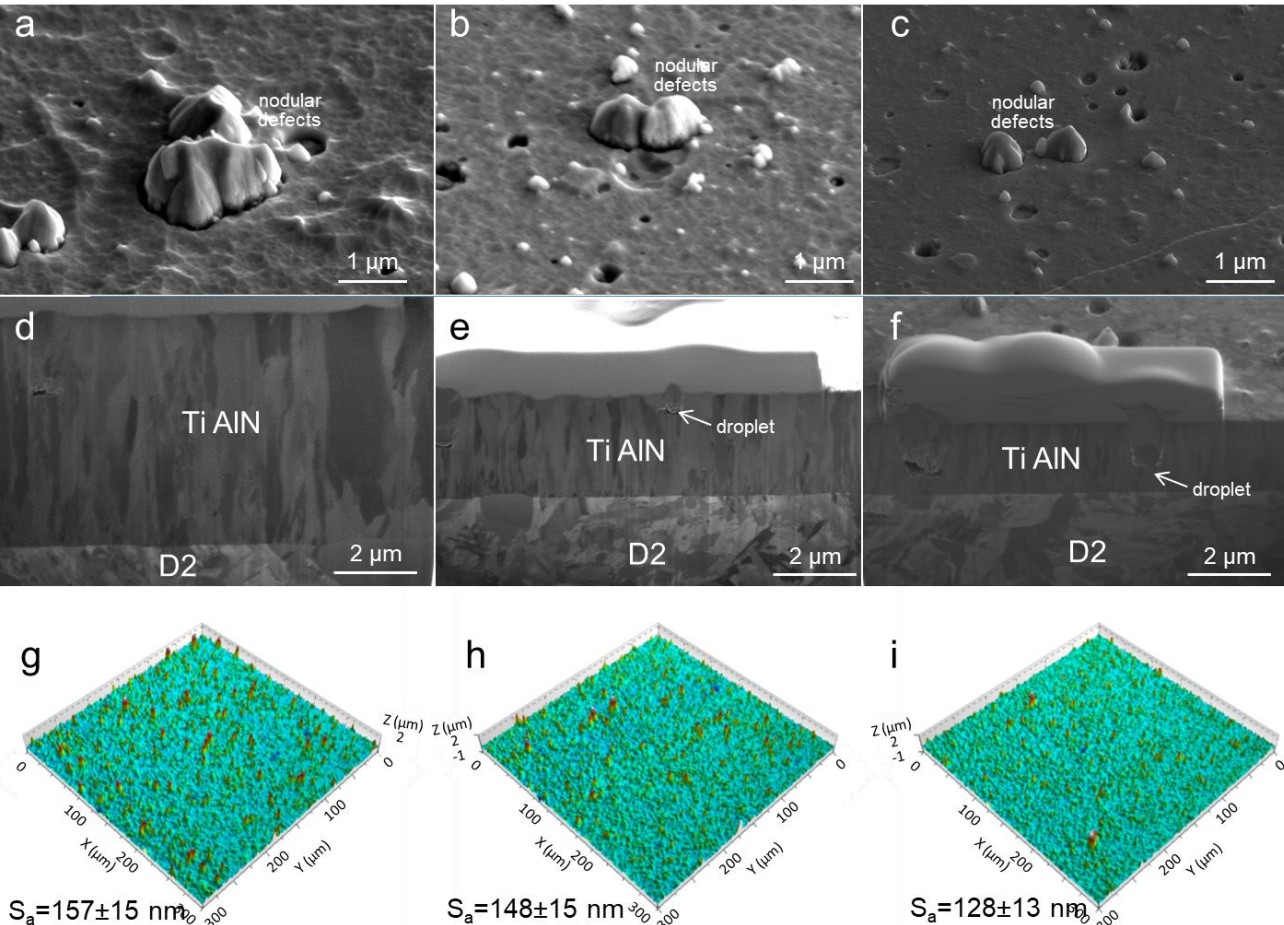

**Figure 28.** Top-view SEM images (**a–c**), SEM images of FIB cross-section (**d–e**) and 3D surface profile images (**g–i**) of TiAlN hard coatings deposited on D2 tool steel substrate in the AIP deposition system using 1-fold (**a,d,g**), 2-fold (**b,e,h**) and 3-fold rotation (**c,f,i**). The magnification scales are identical. The top SEM images were recorded from the side view at about 45° direction.

### 3.3. Influence of Growth Defects on Coating Topography

The growth of the coating is also strongly affected by small foreign particles (e.g., dust particles, flakes) remaining on the substrate surface after the cleaning procedure and those generated during the etching and deposition. All topographical irregularities on the substrate surface are transferred through the coating and even magnified due to the geometrical shadowing effect. The shadowing effect is a geometric phenomenon related to the line-of-sight impingement of arriving atoms. Therefore, even relatively small imperfections with the size of several tens of nanometers cause the growth of micrometer-sized imperfections (growth defects) on the coating surface (Figure 29). Growth defects are generally both larger and more protruding than the carbide-induced coating topography. Growth defects of various shapes and sizes are unevenly distributed, with low numbers and low density, while their size varies from one defect to another (depends on the size and geometry of a seed). Recently, we published a review paper where the reader can find more details related to the growth defects in PVD coatings [17].

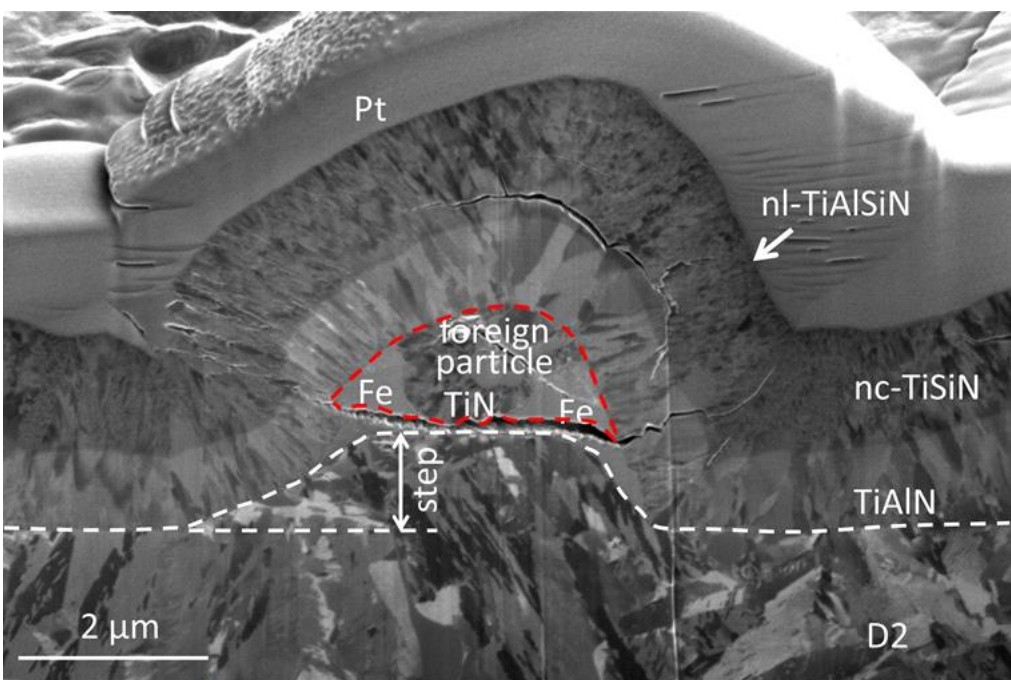

**Figure 29.** The SEM image was taken at the FIB cross-section of the nodular defect in a nanostructured TiAlN/TiSiN/TiAlSiN hard coating. The nodular defect was formed at the site of a foreign particle. A step on the substrate surface beneath this particle proves that it was on the substrate surface even before etching. The substrate beneath the foreign particle is about 0.8 μm thicker than surrounding area, exposed to ion etching.

Growth defects cause a significant increase in the surface roughness. In the area between the growth defects, the coating exhibits a relatively smooth surface with a roughness that is only slightly larger than that of the uncoated substrate. All AFM images of coated substrates presented in this study were taken at the area between the growth defects. Consequently, the AFM surface roughness is much lower than that obtained by the 3D profilometry on a much larger scanning area (Figure 24).

## 4. Conclusions

In this study, we analyzed the surface topography of different PVD hard coatings prepared in four batch-type industrial deposition systems, which differ significantly in both the method of ion etching and deposition. We showed that the coating topography originates from the topography of the substrate surface, intrinsic coating micro-topography and growth defects forming during the deposition process. The substrate surface topography is affected by mechanical pretreatment and in particular by the ion etching method. We found that the etching efficiency depends not only on the etching method and etching parameters but also on the batching configuration and substrate geometry. It is challenging to obtain uniform and controlled ion etching over a large surface because the etching rate varies spatially due to the different batching configurations and complex substrate geometry but also temporally due to multiple rotations of substrates. We also discussed in more detail the influence of ion bombardment during the etching procedure on the surface micro-roughening and the formation of various micro-sized surface features (e.g., cones, pits, steps, trenches).

Hard coatings prepared by various deposition techniques and at different discharge parameters exhibited a wide variety of micro-topographies. The coating morphology is determined by the substrate surface roughness (due to geometrical shadowing effect) and the surface mobility of the depositing atoms. There were considerable differences between the coatings deposited in the same batch using different substrate rotation modes. Specific

rotation mode determines the ratio between the deposition rate and the ion current density to the substrate. Additionally, in the case of two- and threefold rotations, the orientation of the substrate changed significantly and non-periodically, which strongly affected the ion etching and coating deposition. In general, the intensive ion bombardment of the growing coating disrupted its columnar microstructure and changes its topography. Based on these observations, the use of the same deposition protocol for deposition of PVD coatings on the one-, two- and threefold rotating substrates in the same batch is not suitable; a rotation-specific protocol offers a better solution. Batching configuration is also of great importance, both in terms of ion etching and coating deposition. One should avoid batching tools of very different sizes and geometries in the same deposition process.

The biggest changes in the topography of the surface of the coatings were caused by the growth defects. Investigation of the as-deposited coatings showed that there was a significant amount of surface growth defects, mainly nodular defects and shallow craters, both formed during the coating deposition process. These growth defects contributed the most to the roughness of the substrate.

Surface topography analysis of PVD coatings prepared in different industrial deposition systems is a necessary tool to better understand of the relationship between the coating topography and the process parameters. Such analysis should be performed regularly to produce coatings with reproducible performance.

**Author Contributions:** Conceptualization, manuscript writing, P.P.; AFM, SEM and FIB analysis, manuscript review, A.D.; manuscript review, SEM analysis, M.Č.; AFM and TEM analysis, manuscript writing, M.P.; analysis, manuscript writing, SRIM simulation, N.M. All authors have read and agreed to the published version of the manuscript.

**Funding:** This work was supported by Slovenian Research Agency (program P2-0082). We also acknowledge funding from the European Regional Development Funds (CENN Nanocenter, OP13.1.1.2.02.006) and European Union Seventh Framework Programme under grant agreement 312483–ESTEEM2 (Integrated Infrastructure Initiative—I3).

**Institutional Review Board Statement:** Not applicable.

**Informed Consent Statement:** Not applicable.

**Data Availability Statement:** Not applicable.

**Acknowledgments:** The authors would also like to thank Peter Gselman, Tonica Bončina and Gregor Kapun for SEM analyses and Jožko Fišer for performing some laboratory tests.

**Conflicts of Interest:** The authors declare no conflict of interest.

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
