# Peer review of "Surface Topography of PVD Hard Coatings"

_coatings, doi:10.3390/coatings11111387_

Round 1

Reviewer 1 Report

I would like to congratulate the authors for the excellent work presented.
The summary reflects the objective of the work.
The work clearly defines that we are facing an article on surface topography using thin film deposition techniques. It is very well structured and is the result of a detailed investigation that brings value to the scientific community.
The introduction provides a good scientific basis and also provides readers with the importance of roughness as well as the sliding and adhesion phenomena, among others, which contribute to the optimal tribological performance of floor coverings. However, I would like to see more recent articles and work already done on surface analysis by 3D profilometry mentioned.
As an example: https://doi.org/10.1016/j.wear.2021.203730
Note that in 38 articles only 12 are under or 10 years old.
Other recent examples:
https://doi.org/10.3390/ma14185122
 https://doi.org/10.1166/jnn.2012.6760
https://doi.org/10.3390/coatings8110402

In the methods, the coating deposition process is very well characterized and detailed. However, there are questions I would like to ask.
Why didn't they use 3D optical profilometry? Authors could have used to compare with AFM results. For the characterization of the 3D surface, why did the authors not analyze: the amplitude parameters, Sz (Maximum Topographic Surface Height) and Ssk (Skewness of Topography Height Distribution), in addition to the arithmetic, mean of the surface roughness, Sa.
Please indicate the purity of the gases used (Ar, Kr).
It was necessary to characterize the Profilometry roughness analysis.
In the characterization of the samples, it is necessary to mention the number and size of the samples (substrates) as well as their preparation and cleaning, eg cleaning time. After deposition, how were the samples prepared for analysis, what care was taken to ensure the integrity of the samples, for example.
The results are well discussed and extensive, where authors use articles to build on the results. Please enter the standard deviation of the measurements.
The conclusions reflect the analysis of the research and are related
d for the rest of the article.

I would just like to make a repair, in English, the values ​​are separated by a period and not by commas. Example Figure 2: Sa=1,3 nm should be Sa=1.3 nm. please correct this situation throughout the text and pay attention to decimal places. 

Reviewer 2 Report

The in-hand manuscript investigates the surface topography of hard coatings fabricated by three different physical vapor deposition methods, namely low voltage electron beam evaporation, unbalanced magnetron sputtering and cathodic arc evaporation. Overall, the manuscript is very rich with useful findings and what really got my attention is that the authors have prepared their coatings using industrial deposition systems. Such approach and types of study is really needed in the scientific community as it smoothens the transfer of knowledge to the industry/market, something hardly achieved with lab scale systems. Furthermore, the manuscript is very well written, and findings are very promising and well discussed. I highly support the authors work after the approval of the respected editor. I must point out that my only remark on the manuscript is that the Materials and Methods Section needs to be divided into sub-sections, which can easily be done. The authors have done so for the Results and Discussion Section but not for the Materials and Methods Section, which I don’t understand why. At the moment the Materials and Methods Section looks like a bulk of text filled with details. This can easily confuse the reader. The authors may also want to revise the English grammar. Other than that, well done on the great work.

All the best.             

Reviewer 3 Report

Dear authors,

Thank you for submitting the manuscript. I am impressed by the quantity and quality of your work. However, some issues were discovered as I went through the manuscript. Once they are addressed, I believe that it can be an amazing article.

  1. One major issue is the lack of proper description in Figure 3. Four AFM images with line profiles are presented. However, there are no description of the corresponding etching method for each individual AFM image, leaving the reader in confusion, especially when you present five etching methods in Fig. 3a. Although for ASP substrates, only four methods were applied, but then you fail to indicate if ASP or D2 substrates were used for Fig. 3b to 3i. Please add proper description.
  2. Another major issue is with the SEM images in Figure 5 and 6. According to the scale bar, all SEM images covered a relatively small area, about 6.5 µm * 4.5 µm. And although these are all high-quality images, they fail to represent the overall morphology of the sample surfaces. For example, Fig. 6c focused on a small feature (an exotic particle maybe?), instead of presenting the smooth surface of H11 substrate. I believe that AFM images have enough resolution to reveal small features, and SEM images can be utilized to present an overview of the topography. A better example has actually been given in Fig. 8b, which covers a larger area and gives me a clear overview of the surface morphology. Please consider adding SEM images in Fig. 5 and 6 to cover a larger area and provide an overview like Fig. 8b, and use the current ones as an insertion. It can be challenging for samples with very smooth surfaces, but I believe it’s achievable.
  3. In Fig. 27, line profiles are not provided, but there are lines in the middle of all AFM images indicating where the line profiles are likely taken from. Is there anything missing? If you did not intend to include line profiles here, please remove the lines in AFM images. Personally, I think it’s better to provide line profiles for consistency, as you provided line profiles for other AFM images (Fig. 2, 3,5 and 6). Also, the z-scale of Fig. 27a is significantly larger compared to Fig. 27b and 27c because of large surface roughness, adding line profiles here can present this difference more clearly.
  4. Also, in Fig. 27a, there is a white spot in the upper part of the image indicating a sudden jump in height that seems not natural. What caused it? Is it a surface defect or an unexpected instrument drift?
  5. In Line 324, you used Fig. 3 to support the claim that etching efficiency can be affected by etching methods. However, for different etching methods, you used different etching time as well. In my opinion, Fig. 3 cannot support your claim, controlled etching time for different etching methods are needed. If supplement experiment is not achievable, you need to rewrite this sentence to make it less misleading.
  6. Two paragraphs (Line 441 to 461 and Line 462 to 479) used Fig. 15 as evidence. I think in both paragraphs, you were discussing the batch configuration and its impact on etching configuration. However, the detailed description of Fig. 15 was put in the second paragraph (Line 467 to 472) instead of the first one. Please move it to the first paragraph where you mentioned Fig. 15 for the first time. Or you may also consider integrating these two paragraphs into one.
  7. In Table 1, the etching time, working pressure and substrate bias of the deposition system CC800/9 do not match with your description in the article (Line 189 and 198). What are the actual parameters? Please make corrections accordingly.
  8. In Fig. 24, were these TiAlN coatings deposited with intermediate etching? Why the coating on H11 substrate has a much smaller surface roughness? Is it because of the initial ion etching? This is not a major issue, just a genuine question.
  9. The introduction part of the article is very comprehensive and well-structured, but I believe some additional references could be helpful. For example, from Line 76 to Line 96 the paragraph used only 3 references. Some of the sentences (Line 78 to 84) in the paragraph is not common sense to me, some additional references could be useful.

Thank you!

Reviewer 4 Report

Purpose of paper is on the topography of PVD hard coatings deposited on different substrates by various PVD techniques.  Authors focused on differences between PVD methods both in the ion etching and the deposition steps. The analysis coating preparation factors that can affect the topography. The paper is important for the understanding the surface density and artefacts of the growth defects.  The hypothesis were confirmed analyses of coatings prepared by various deposition techniques. The coating morphology was described properly by determination of the substrate surface roughness and the surface mobility of the depositing atoms.  The changes in the topography of the surface of the coatings were caused by the growth defects. The growth defects contribute to the roughness of the substrate. Surface topography analysis of PVD coatings prepared in different industrial deposition systems is a necessary tool to better understand of relationships between the coating topography and the process parameters. The hypothesis were confirmed by the adequate Surface analysis techniques such as AFM, SEM and  STEM. The methodology is properly chosen and  results described properly. The novelty is on understanding the surface density and artefacts of the growth defects.  The novelty is sufficient for publication in Coatings.

Round 2

Reviewer 3 Report

In this manuscript, the authors focused on the topography of PVD coatings and how it could be affected by substrate pre-treatment, especially ion etching, and deposition parameters. The highlight of the article is the utilization of multiple industrial deposition systems, and the study of batch configuration and complicated substrate geometry and how they might affect the surface morphology. Overall, the article is in high quality, and the authors have addressed properly the concerns I had during the first review.